

# Evaluating a fire smoke simulation
# algorithm in the National Air Quality
# Forecast Capability (NAQFC) by using
# multiple observation data sets during the
# Southeast Nexus (SENEX) field campaign
Li Pan [1,2], Hyun Cheol Kim [1,2], Pius Lee [1], Rick Saylor [3], YouHua Tang[1,2], Daniel Tong [1,2], Barry Baker[1,2],
Shobha Kondragunta [4], Chuanyu Xu [5], Mark G. Ruminski [4], Weiwei Chen [1,6], Jeff Mcqueen [7] and Ivanka
Stajner[8]
[1] NOAA/OAR/Air Resources Laboratory, College Park, MD 20740, USA
[2] UMD/Cooperative Institute for Climate and Satellites, College Park, MD 20740, USA
[3] NOAA/OAR/ARL/Atmospheric Turbulence and Diffusion Division, Oak Ridge, TN 37830, USA
[4] NOAA/NESDIS, College Park, MD 20740, USA
[5] I. M. Systems Group at NOAA, College Park, MD 20740, USA
[6] Northeast Institutes of Geography and Agroecology, Chinese Academy of Sciences, Changchun 130102,
P. R. China
[7] NOAA/NCEP/Environmental Modeling Center, College Park, MD 20740, USA
[8] NOAA/NWS Office of Science and Technology Integration, Silver Spring, MD 20910, USA
Correspondence to: Li.Pan@noaa.gov



## Abstract

Multiple observation data sets: Interagency Monitoring of Protected Visual Environments
(IMPROVE) network data, Automated Smoke Detection and Tracking Algorithm (ASDTA), Hazard
Mapping System (HMS) smoke plume shapefiles and aircraft acetonitrile ($CH_3CN$) measurements from
the NOAA Southeast Nexus (SENEX) field campaign are used to evaluate the HMS-BlueSky-SMOKE-
CMAQ fire emissions and smoke plume prediction system.  A similar configuration is used in the US
National Air Quality Forecasting Capability (NAQFC).  The system was found to capture most of the
observed fire signals. Usage of HMS-detected fire hotspots and smoke plume information were valuable
for both deriving fire emissions and forecast evaluation. This study also helped identified that the
operational NAQFC did not include fire contributions through lateral boundary conditions resulting in
significant simulation uncertainties.  In this study we focused both on system evaluation and evaluation
methods. We discussed how to use observational data correctly to filter out fire signals and
synergistically use multiple data sets.  We also addressed the limitations of each of the observation data
sets and evaluation methods.

## Introduction

Wildfires and agricultural/prescribed burns are common in North America all year round, but
predominantly occur during the spring and summer months (Wiedinmyer et al., 2006). These fires pose
a significant risk to air quality and human health (Delfino et al., 2009; Rappold et al., 2011; Dreessen et
al., 2016; Wotawa and Trainer 2000; Sapkota et al., 2005; Jaffe et al., 2013; Johnston et al., 2012). Since
January 2015, smoke emissions from fires have been included in the National Air Quality Forecasting
Capability (NAQFC) daily $PM_{2.5}$ operational forecast (Lee et al., 2017). The NAQFC fire simulation consists
of: the NOAA National Environmental and Satellite Data and Information Service (NESDIS) Hazard
Mapping System (HMS) fire detection algorithm, the U.S. Forest Service (USFS) BlueSky-fire emissions





estimation algorithm, the U.S. EPA Sparse Matrix operator Kernel Emission (SMOKE) applied for fire
plume rise calculations, the NOAA National Weather Service (NWS) North American Multi-scale Model
(NAM) for meteorological prediction and the U.S. EPA Community Multi-scale Air Quality Model (CMAQ)
for chemical transport and transformation.  In contrast to most anthropogenic emissions, smoke
emissions from fires are largely uncontrolled, transient and unpredictable. Consequently, it is a
challenge for air quality forecasting systems such as NAQFC to describe fire emissions and their impact
on air quality (Pavlovic et al., 2016; Lee et al., 2017; Huang et al., 2017).

Southeast Nexus (SENEX) was a NOAA field study conducted in the Southeast U.S. in June and

July 2013 (Warneke et al., 2016). This field experiment investigated the interactions between natural
and anthropogenic emissions and their impact on air quality and climate change (Xu et al., 2016;
Neuman et al., 2016). In this work, we used the SENEX dataset to evaluate the HMS-BlueSky-SMOKE-
CMAQ fire simulations during the campaign period.

Two simulations were performed: one with and one without smoke emissions from fires during

the SENEX field campaign. Due to the large uncertainties in the estimates of fire emissions and smoke
simulations (Baker et al., 2016; Davis et al., 2015; Drury et al., 2014), the first step of the evaluation
focused on the fire signal capturing capability of the system. Differences between the two simulations
represented the impact of the smoke emissions from fires on the CMAQ model results. Observations
from various sources were utilized in this analysis:  (i) ground observations (Interagency Monitoring of
Protected Visual Environments (IMPROVE)), (ii) satellite retrievals (Automated Smoke Detection and
Tracking Algorithm (ASDTA) and HMS smoke plume shape), and (iii) aircraft measurements (SENEX
campaign). Fire signals predicted by the modeling system were directly compared to these observations.
Several criteria have been used to rank efficacy of the observation systems for fire induced pollution
plumes.



## Methodology

In this section we introduce the NAQFC fire modeling system used in the study. Uncertainties
and limitations in the various modeling components of the system are discussed. Fig. 1 illustrates the
schematics of the system. There are four processing steps:

## HMS (Hazard Mapping System)

The NOAA NESDIS HMS is a fire smoke detection system based on satellite retrievals. The
satellite constellation used comprised of 2 Geostationary Operational Environmental Satellite (GOES-10
and GOES-12) and 5 polar orbiting satellites (MODIS (Moderate-resolution Imaging Spectroradiometer))
-- Terra and Aqua, AVHRR (Advanced Very High Resolution Radiometer) 15/17/18).  HMS detects
wildland fire locations and analyzes their sizes, starting times and durations (Ruminski et al., 2008;
Schroeder et al., 2008; Ruminski and Kondragunta 2006).
HMS first processes satellite data by using automated algorithms for each of satellite platforms
to detect fire locations (Justice et al., 2002; Giglio et al., 2003; Prins and Menzel 1992; Li et al., 2000),
which is then manually analyzed by analysts to eliminate false detections and/or add missed fire
hotspots. The size of the fire is represented by the number of detecting pixels corresponding to the
nominal resolution of MODIS or AVHRR data.  Fire starting times and durations are estimated from close
inspection of the visible band satellite imagery.  A bookkeeping file is generated at the end of this
detection step, named "hms.txt". It includes all the thermal signal hotspots detected by the
aforementioned 7 satellites. During the analyst quality control step, detected potential fire hotspots
lacking visible smoke in the retrieval's RGB real-color imagery are removed resulting in a reduced fire
hotspot file called either "hmshysplit.prelim.txt" or "hmshysplit.txt" to be input into the BlueSky
processing step.



In general, "hmshysplit.prelim.txt" and "hmshysplit.txt" are very similar, and "hmshysplit.txt" is

created later than "hmshysplit.prelim.txt" (Fig. 1). But the differences between "hmx.txt" and

"hmshysplit.txt" ("hmshysplit.prelim.txt") can be rather substantial. The reasons for differences are: 1)

many detected fires do not produce detectable smoke; 2) some fires/hotspots are detected only at

night, when smoke detection is not possible; 3) smoke emission RGB imageries are obscured by clouds

thus not detected by the analyst. Therefore, smoke emission occurrence provided by the HMS is a

conservative estimate of fire emissions.

By using multiple satellites the likelihood of detecting fires in HMS is robust. However, when the

fire geographical size is small the HMS detection accuracy dramatically decreases (Zhang et al., 2011; Hu

et al., 2016).  Other limitations of the HMS fire detections include ineffective retrievals at nighttime and

under cloud cover.

**BlueSky**

BlueSky, developed by the USFS (US Forest Service), is a modeling framework to simulate smoke

impacts on regional air quality (Larkin et al., 2009; Strand et al., 2012). In this study, BlueSky acted as a

fire emission model to provide input for SMOKE (Herron-Thorpe et al., 2014; Baker et al., 2016). BlueSky

calculates fire emission based on HMS-derived locations (Fig. 1).

Fire geographical extent is reflected by the number of nearby fire pixels detected by satellites in

a 12 km resolution CMAQ model grid. Fire pixels are converted to fire burning areas in BlueSky based on

the assumption that each fire pixel has a size of 1 km$^2$ and 10% of its area can be considered as burn-

active (Rolph et al., 2009). All fire pixels in a 12 km grid square are aggregated. BlueSky uses the

following to estimate biomass availability: fuel loading map is from the US National Fire Danger Rating

System (NFDRS) for the Conterminous US (CONUS) with the exception in western US where the HARDY

set is used (Hardy and Hardy 2007). BlueSky uses Emissions Production Model (EPM) (Sandberg and



Peterson 1984), a simple version of CONSUME, to calculate fuel actually burned -- the so-called
consumption sums. Finally, EPM is also used in BlueSky to calculate the fire emission hourly rate per
grid-cell. BlueSky outputs CO, $CO_2$, $CH_4$, non-methane hydrocarbons (NMHC), total PM, $PM_{2.5}$, $PM_{10}$ and
heat flux (Fig. 1).

BlueSky does not iteratively recalculate fire duration according to the modeled diminishing fuel

loading or the modeled fire behavior.  In the aggregation process, when there is more than one HMS
point in a grid cell which have different durations, all points in that grid cell would be assigned the
largest duration in all points. For an example, if there were 3 HMS points that had durations of 10, 10
and 24 hours, the aggregation would include 3 points (representing 3 $km^2$) assigned with 24 hour
duration to all of the 3 HMS points.

HMS has no information about fuel loading. BlueSky uses a default fuel loading climatology over

the eastern US. BlueSky uses an idealized diurnal profile for fire emissions. Uncertainties in fire sizes,
fuel loading and fire emissions rate lead to large uncertainties in wildland smoke emissions (Knorr et al.,
2012; Drury et al., 2014; Davis et al., 2015).
**SMOKE**

In SMOKE (Sparse Matrix Operator Kernel Emission), the BlueSky fire emissions data in a

longitude-latitude map projection are converted to CMAQ ready emission gridded files (Fig. 1).  Fire
smoke plume rise is calculated using formulas by Briggs. The heat flux from BlueSky and NAM
meteorological state variables are used as input (Erbrink 1994). The Briggs' algorithm calculates plume
top and plume bottom, between plume top and bottom the emission fraction is calculated layer by layer
assuming a linear distribution of flux strength in atmospheric pressure. For model layers below the
plume bottom the emission fraction is assumed to be entirely in the smoldering condition as a function
of the fire burning area.





We adopted a speciation cross-reference map to match BlueSky chemical species to that in
CMAQ using the U.S. EPA Source Classification Codes (SCCs) for forest Wildfires
(https://ofmpub.epa.gov/sccsearch/docs/SCC-IntroToSCCs.pdf). The life-span of fire is based on the HMS
detected fire starting time and duration. During fire burning hours a constant emission rate is assumed.
This constant burn-rate has been shown to be a crude estimate (Saide et al., 2015; Alvarado et al.,
2015). Other uncertainties include plume rise (Sofiev et al., 2012; Urbanski et al., 2014; Achtemeier et
al., 2011) and fire-weather (fire influencing local weather).
**CMAQ**
The CMAQ version 4.7.1 was used. We chose the CB05 gas phase chemical mechanism (Yarwood
et al., 2005) and the AERO5 aerosol module (Carlton et al., 2010). Anthropogenic emissions were based
on the U.S. EPA 2005 National Emission Inventory (NEI) projected to 2013 (Pan et al., 2014), Biogenic
emissions (BEIS 3.14) were calculated in-line inside CMAQ.
**Simulations**
The NAM provided meteorology fields to drive CMAQ (Chai et al., 2013). NAM meteorology is
evaluated daily and results (BIAS and RMSE etc.) are posted on:
"http://www.emc.ncep.noaa.gov/mmb/mmbpll/mmbverif/". The simulation domain is shown in Fig. 1.
It includes two domains: (i) a 12km domain covering the Continental U.S. (CONUS); and (ii) a 4km
domain covering the Southeast U. S. where the majority of SENEX measurements occurred.  Lateral
boundary conditions (LBC) used in the smaller SENEX domain simulation were extracted from that from
the CONUS simulations. Four scenarios were simulated: CONUS with fire emissions, CONUS without fire
emissions, SENEX with fire emissions and SENEX without fire emissions.
There were several differences in system configuration between the NAQFC fire smoke
forecasting and the "with-fire" simulation in this study. For models, the BlueSky versions used in NAQFC





and that in this study are v3.5.1 and v2.5, respectively; CMAQ versions used in NAQFC and in this study
are v5.0.2 and v4.7.1, respectively.  For simulations, current fire smoke forecasting in the NAQFC
includes two runs: the analysis and the forecast (Huang et al. 2018 (manuscript in preparation)). The
analysis run is a 24-hour retrospective simulation to rerun yesterday's fire emissions using yesterday's
meteorology providing an initial condition for today's forecast. The forecast run is a 48-hour forecasting
simulation to run yesterday's fire emissions to be projected as continued fires had their burn duration
were more than 24 hours.  The "with-fire" simulation in this study is exactly identical to the analysis run
in NAQFC.
**Evaluations**

Carbon monoxide (CO) has a relatively long life time in the air and is commonly associated with

biomass burning.  CO was used as a fire tracer in the prediction. The CO difference (ΔCO) between
CMAQ simulations with and without fire emissions was used as the indicator of fire influence. For
additional observations we used: potassium (K) collected at the IMPROVE (Interagency Monitoring of
Protected Visual Environments) sites within the SENEX domain; acetonitrile ($CH_3CN$) measured from the
SENEX campaign flights; and fire plume shape detected by the HMS analysis as real fire signals.  The
enhancement in ΔCO concentration due to fire was directly compared with those signals. At the same
time, ΔAOD (Aerosol Optical Depth) from CMAQ ("with-fire" simulated concentration minus that with
"without-fire") was also used as fire indicator when compared with smoke masks given by the ASDTA
(Automated Smoke Detection and Tracking Algorithm).

In this study, we have focused on evaluations subject to the large uncertainties of the underlying

physical processes of smoke emissions from fires and its transport. In each modeling step in HMS,
BlueSky, SMOKE and CMAQ, the modeling system accrues uncertainties. Such uncertainties  were likely
cumulative and might lead to larger error in succeeding components (Wiedinmyer et al., 2011). For an



example, heat flux from BlueSky influenced plume rise height in SMOKE and consequently influenced
plume transport in CMAQ. It is also noteworthy that when we compared modeled ΔCO against
measured K or $CH_3CN$, the objective was to search for enhancement signals resulting from fires but it
was not aiming to account for proportional concentration changes in the tracers.  Attempting to account
for CMAQ simulation uncertainties in surface ozone and particulate matter as a function of smoke
emissions from fires was difficult.  Neither was it the objective of this study.  Rather, the purpose of this
study is to focus on analyzing the capability of the HMS-BlueSky-SMOKE-CMAQ modeling system to
capture the timing of fire signals.

The SENEX campaign occurred in June and July and our model simulations were from June 10 to

July 20, 2013. Throughout the campaign we used all available observation datasets including ground-,
air- and satellite-based acquired data. Each dataset had its unique characteristics and linking them
together gave an overall evaluation. At the same time, in each dataset our evaluations included as many
as possible observed fire cases.  Both well-predicted and poorly-predicted cases are presented to
illustrate potential reasons responsible for the modeling system's behavior.

## Results and Discussions

### Observed CO versus modeled CO in SENEX

Table 1 lists observed and modeled CO vertical profiles for the "with-fire" and "without-fire"

cases during the SENEX campaign. Observed CO concentrations between the surface and 7 km AGL
(Altitude above Ground Level) in the SENEX domain area remained greater than 100 ppb during all 40
days of the campaign. The highest CO concentrations were measured closer to the surface. The
maximum measured CO concentration of 1277 ppb was observed during a flight on July 03 at an ASL
(Altitude above Sea Level) of 974 m. In this flight strong fire signals were observed but the fire
simulation system missed those signals as discussed below.



CO concentrations were underestimated by the model in almost all cases even when the model
captured CO contribution from fire emissions spatio-temporarily.  Mean ΔCO in each height interval was
usually above 1.5 ppb but less than 2.0 ppb. Fig. 2a exhibits the contribution of total CO emissions from
fires which occurred inside the SENEX domain over the simulation period. The maximum CO emissions
contribution from fires was about 3% during the campaign. In most of those days fire emission
contributions in SENEX were less than 1%. The averaged contribution during those 40 days was 0.7%.
Fig. 2b exhibits the contribution of CO flowing into the SENEX domain from its boundary caused by fire
outside the SENEX domain but inside the CONUS domain (Fig. 1). The averaged fire contribution to CO
from outside the SENEX domain was 0.67%. CO influenced by fire emission in June is greater than that in
July.
During the field experiment the general lack of large fires made evaluation of modeled fire
signature difficult since it was easier to capture large fire signals than the smaller fires. We postulated
that a clear fire signal simulated in the HMS-BlueSky-SMOKE-CMAQ system could be indicated by ΔCO
significantly larger than its temporal averages resulted by fires originated from inside and/or outside the
SENEX domain. For an example, if a clear fire signal between 500 m and 1000 m AGL is represented by
ΔCO in a model simulation and the concentration of ΔCO is above 2.0 ppb, based on the average CO
concentration of about 150 ppb as well as on with SENEX domain and outside of SENEX domain fire
contributions of (150*(0.007+0.0067) =2.0).
Figure 3 displays the simulated ΔCO extracted along a SENEX flight path. The modeled
concentration showed that the fire impacts on SENEX were not negligible perspective despite a lack of
larger fire events as shown in Fig. 2a and 2b during the SENEX campaign period. That confirmed the
importance of evaluating the fire simulation system in an air quality model. Unless a model is able to
predict fire signals correctly it is useless for modelers to discuss fire effects on chemical composition of




the atmosphere. A detail of how our model caught or missed or falsely predicted fire signals during the
SENEX campaign and a comparison of ΔCO versus CH$_3$CN will be discussed in the follow discussion.
**IMPROVE**

The Interagency Monitoring of Protected Visual Environments (IMPROVE) is a long term air

visibility monitoring program initiated in 1985 (http://vista.cira.colostate.edu/Improve/data-page). It
provides 24-h integrated particulate matter (PM) speciation measurements every third day (Malm et al.,
2004; Eatough et al., 1996). The IMPROVE dataset was chosen for this analysis because it included K
(potassium), OC (organic carbon) and EC (elemental carbon), important fire tracers. IMPROVE monitors
are ground observation sites likely influenced by nearby fire sources.

There were 14 IMPROVE sites in the SENEX domain (Fig. 4).  Potential fire signals were identified

by using CMAQ modeled ΔCO and IMPROVE observed K. However, in addition to fires K has multiple
sources such as soil, sea salt and industry. Co-incidentally fires should also produce enhanced EC and OC
concentrations, a fire signal should reflect above-average values for EC, OC, and K.  EC, OC and K
observations that were 20% above their temporal averages during the SENEX campaign were used as a
predictor for fire event identification.  Meanwhile, co-measured NO$_3^-$ and SO$_4^{2-}$ concentrations 50%
below their respective temporal averages was used to screen out data with industrial influences. Lastly,
a third predictor was employed so that concentrations of other soil components besides K should be
below their temporal average to eliminate conditions of spikes in K concentration due to dust. With
these three criteria the IMPROVE data was screened for fire events (See Table 2).

Five fire events were observed at four IMPROVE sites. Table 2 lists measured EC, OC, NO$_3^-$, K, soil

and SO$_4^{2-}$ concentrations (μg m$^{-3}$) and their ratios to averages. BC versus OC and K versus BC ratios were
also calculated and listed in Tab. 2 to illustrate the application of our criteria. We found that except for
monitor BRIS, all other sites (COHU, MACA and GRSM) had BC/OC and K/BC ratios comparable to the



ratios of the same quantities due to biomass burning reported by other researchers (Reid et al., 2005;
DeBell et al., 2004).  BRIS is a coastal site likely influenced by sea salts (Fig. 4).

For the four identified fire cases, we plotted ΔCO as a modeled fire tracer around the IMPROVE

sites. Our model simulation reproduced fire signals on June 21 at COHU and GRSM and on June 24 at
MACA. We used the June 24 MACA case as an example (see Fig. 4) -- closed black circles represent the
detected fire locations; closed triangles represents IMPROVE sites, and ΔCO values above 2.0 ppb were
shown. On June 24, 2013, detected fire spots were outside the SENEX domain, but SSW wind blew
smoke plumes into the SENEX domain and affected modeled CO in MACA. Modeled ΔCO in MACA was 5
ppb.

Another IMPROVE site located upwind of MACA, CADI, was also potentially under the influence

of that fire event; however, data from CADI on June 24 did not indicate a fire influence, possibly due to
the frequency of IMPROVE sampling that eluded measurement or that the smoke plume was
transported above the surface in disagreement with what was modeled.  Within the four fire cases
identified by the IMPROVE data during SENEX (Tab. 2), the model successfully captured three out of four
events. The model missed fire signal on July 3 at MACA. The model missed the fire signal on July 3 at
MACA. The following section is dedicated for the July 3 SENEX flight.
**Plume Spatial Coverage**

HMS determines fire hotspot locations associated with smoke and upon incorporating the

smoke plume shape information from visible satellite images. HMS provides smoke plume shapefiles
over much of North America. We focused on the shapefile over CONUS – a two-dimensional smoke
plume spatial depiction collapsing all plume stratifications to a satellite eye-view. For modeled plumes,
we integrated modeled ΔCO by multiplying the layer values with the corresponding CMAQ model layer



thicknesses and air density to derive a simulated smoke plume shape. HMS-derived smoke plume shape
versus CMAQ predicted smoke plume shape was then used to evaluate the fire simulation.

Figure of Merits in Space (FMS) (Rolph et al., 2009) is a statistic for spatial analysis and was

calculated as follows:

$$FMS = \frac{Area\_hms \ \cap \ Area\_cmaq}{Area\_hms \ \cup \ Area\_cmaq} \ X \ 100\%$$

Where Area_hms represents area of grid cells influenced by fire emission over CONUS detected by HMS
and Area_cmaq represents area of grid cells over CONUS identified by model prediction.  In general, a
higher FMS value indicates a better agreement between the observed and modeled plume shape (Rolph
et al., 2009).

Figure 5 summarizes FMS during the SENEX campaign.  Average FMS was 22% with its maximum

at 56% on July 6 and minimum at 1.2% on June 17 2013. Figure 6a exhibits HMS detected smoke plume
and CMAQ calculated smoke plume over CONUS on July 6. The light blue shading represents modeled
plume shape (defined as total column ΔCO) and the thin dash line and emboldened green lines encircle
areas representing HMS-derived light and strong influenced plume shape, respectively (Fig. 6a). The FMS
score was 56% meaning that the modeled plume shape was consistent with that of HMS. However,
CMAQ might have underestimated the intensive fire influence areas along the border of California and
Nevada. Subsequently, the model also under-predicted its associated influence in North Dakota, South
Dakota, Minnesota, Iowa and Wisconsin.

Figure 6b exhibits the worst case on June 17 2013 in terms of resulting with a FMS score at 1.2%.

Two reasons led to this: (i) CMAQ missed fire emissions from Canada. Those fire sources located outside
the CONUS modeling domain and our simulation system used a climatologically-based static LBC;
Secondly on June 17, there were a lot of fire hotspots in the Southeastern U.S., i.e., in Louisiana,



Arkansas and Mississippi along the Mississippi River. Hotspots were detected but they lacked associated
smoke in corresponding RGB imagery (Fig. 6c). This could be due to cloud blockage or to small
agricultural debris clearing, burns in under-bushes or prescribed burns.  These conditions prevented the
HMS from identifying fires and hence emissions were not modeled for those sources.

It is noteworthy that the FMS evaluation contained uncertainties contributed from both

modeled and observed values. The calculated campaign duration and SENEX-wide averaged FMS was
22%. It is significantly higher than that achieved by a similar analyses done by HYSPLIT (Hybrid Single
Particle Lagrangian Integrated Trajectory) smoke forecasting for the fire season of 2007 (6.1% to 11.6%)
(Rolph et al., 2009). The primary reason is that the HYSPLIT smoke simulation accessed at the invocation
of a forecast cycle the HMS fire information which is already one day old due to retrieval latency and
cycle-queuing issues. However, our model simulation in this study was from a retrospective module
using current day HMS fire information. Such discrepancies have been discussed by Huang et al. 2018
(*manuscript in preparation*). Other reasons, such as plume rise, etc. were discussed in the following
section on ASDTA.
**ASDTA**

The Automated Smoke Detection and Tracking Algorithm (ASDTA) is a combination of two data

sets: (1) the NOAA Geostationary satellite (G13) retrieves aerosol optical depth using visible channels
and produces a product called GOES Aerosol/Smoke Product (GASP) (Prados et al., 2007); and, (2) the
NOAA HYSPLIT dispersion model predicts smoke plume direction and extension (Draxler and Hess 1998).
ASDTA provides the capability to determine whether the GASP is influenced by one or multiple smoke
plumes over a location at a certain time.  The ASDTA is a signature identification analysis. On the other
hand, the HYSPLIT smoke forecast is based on the HMS fire detection and BlueSky emission modeling
driven by the NOAA NWS regional meteorology model. These data are suitable for model performance



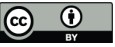

evaluation in this study.  For each simulation, modeled AOD was calculated for each sensitivity test
("with-fire" or "without-fire") and ΔAOD is defined as the difference obtained by subtracting
AOD_without-fire from AOD_with-fire.
Figure 7a illustrates a GOES retrieved AOD (summed over from 10:00 am to 2:00 pm at local
time) contour plot that reflects influences by smoke plumes over the CONUS domain on June 14 2013.
Color-shaded region represents the fire-smoke influenced areas and the color denotes the magnitude of
the retrieved AOD (Fig. 7a).  Figure 7b presents similar results, but for simulated ΔAOD (with-fire −
without-fire).  For further evaluation of the HMS detected smoke plume shape Fig. 7c can be compared
with Figs. 7a and 7b.
Figure 7a shows several regions under the influence of fires in: California, northwest Mexico,
Kansas, Missouri, Oklahoma, Arkansas, Texas and part of the Gulf of Mexico.  In the northeastern USA,
fire plumes occurred sparingly.   Those regions agreed relatively well with the shaded contours between
Figs. 7a and 7c. However, due to the lack of fire treatments in the CMAQ LBC, the simulation (Fig. 7b)
missed smoke influence on the northeast region of the CONUS domain.  CMAQ also failed to simulate
the fire influences in the southwest region of the domain.
Similar plots for June 25 are shown in Figs. 7d, 7e and 7f for ASDTA, CMAQ and HMS,
respectively.  The ASDTA (Fig. 7d) predicted an overestimation in fire influences in the south including
Texas and the Gulf of Mexico and an underestimation in the northeastern U.S.  On the other hand, the
model predicted two strong fire signals clearly: near the border between Arizona and Mexico, and in
Colorado (See Fig. 7e). All the fire influenced areas in Fig. 7e were seen in Fig. 7f --- reflecting
observation by HMS.
Comparing ASDTA plots and CMAQ ΔAOD plots (Fig. 7a vs 7b; Fig. 7d vs 7e), we found both
similarities and differences. Similarities were attributable to similar fire accounting, smoke emissions



from fires calculation and meteorology. Differences were attributable to: (i) HYSPLIT smoke simulation
used more fire hotspots than that used by CMAQ due to domain size; (ii) only fires inside the CONUS
were included in the CMAQ fire simulation and LBCs did not vary to reproduce impacts of wildfires from
outside of the domain; (iii) Despite both the HYSPLIT and CMAQ fire plume rise were estimated by the
Briggs' equation, the HYSPLIT plume rise was limited to 75% of the mixed layer height (MLH) at daytime
and two times MLH at nighttime, whereas the CMAQ fire plume rise did not have these limitations.
**SENEX**
SENEX (Southeast Nexus) was a field campaign conducted by NOAA in cooperation with the US
EPA and the National Science Foundation in June and July 2013. Although SENEX was not specifically
designed for fire studies, its airborne measurements included $PM_{2.5}$ OC and EC, CO and acetonitrile
($CH_3CN$).  $CH_3CN$ was chosen as a fire tracer since it is predominantly emitted from biomass burning
(Holzinger et al., 1999; Singh et al., 2012).
$CH_3CN$ has a residence time in the atmosphere of around 6 months (Hamm and Warneck 1990)
and the reported $CH_3CN$ background concentration is around 100 - 200 ppt (Singh et al., 2003).
Measured $CH_3CN$ concentrations tend to increase with altitude (Singh et al., 2003; de Gouw et al., 2003),
since biomass burning plumes are subject to ascend during long-range transport. During SENEX,
measured $CH_3CN$ showed a similar pattern. Fire signals were identified through airborne measurements
of $CH_3CN$ when its concentration exceeded the background; e.g., on July 3 2013, or when its
concentration peak appeared at high altitude; e.g., on June 16 2013 and July 10 2013.
$CH_3CN$ airborne measurements were used to identify fire plumes at certain locations and
heights during SENEX. For model evaluations, fire locations and accurate meteorological wind field are
crucial to interpret 2-D measurements such as IMPROVE, HMS and ASDTA. To verify a 3-D fire field, it is
critical to capture plume rise. However, it was extremely difficult to back out plume rise from the



airborne measurements. An additional uncertainty arose in the difference of temporal resolutions of the
data: IMPROVE, HMS shapefiles and ASDTA were daily or hourly data, whereas airborne CH₃CN data
were measured at one-minute intervals.

Figure 8a shows a CMAQ simulated ΔCO vertical distribution along flight transects on June 16

2013. The x-axis label is UTC (hour) and the y-axis label is AGL (m). Two color bars represent observed
CH₃CN concentration (rectangle bar in ppt) and simulated ΔCO concentration (fan bar in ppb),
respectively. This flight occurred during the weekend over and around power plants around Atlanta, GA.
The color of flight path represents observed CH₃CN concentration in ppt. In Fig. 8a, the concentration of
ΔCO increased from surface to 5000 m, especially above 2000 m. Six CH₃CN concentration peaks were
observed when AGL was above 2500 m.

For CMAQ simulated ΔCO, five out of six fire signals detected by CH₃CN measured spikes were

captured where ΔCO concentrations were all above 3 ppb. Only one fire signal was missed by the model
at 18:30 UTC June 16 2013. Model simulation showed that long range transports (LRT) of smoke plumes
influenced airborne observations. Fire signals from the free troposphere subsided and influenced flight
measurements. High EC or OC or CO did not concur with high CH₃CN observation probably due to
species lifetime differences.  HMS smoke plume did not show any hotspots or smoke plume around
Atlanta suggesting that the sources of those observed fire signals were not from its vicinity.

A similar phenomenon was seen in SENEX flight 0710, which occurred during flight transects

from Tennessee to Tampa, FL. Figure 8b is a similar graph as Fig. 8a. Based on ΔCO concentrations,
CMAQ captured the July 10 case as fire signals were observed.  Nonetheless, ΔCO may be over predicted
at around 19 UTC. The model exhibited a fire signal with ΔCO concentration of about 3 ppb near 6000 m
around 19 UTC, whereas measured CH₃CN was 120 ppt and decreased with AGL.





## SENEX flight on July 3

Observations from IMPROVE, HMS and SENEX identified fire signals on July 3 2013. ASDTA
retrievals were not available. Those signals were missed by the model. In this section, we will use all of
evaluation methods addressed above to study potential causes of failure of the model to reproduce fire
signals.
At the MACA IMPROVE site on July 3 2013, the wind direction at the surface was southeasterly,
with no fire hotspots (solid black circle) located upwind of MACA (Fig. 9a). Without any identified
hotspots upwind, the model missed fire signals observed at MACA on July 3 2013.
Flight #0703 was a night mission targeting power plants in Missouri and Arkansas. The flight
path is shown in Fig. 9b and is colored by measured $CH_3CN$ concentration. In order to highlight $CH_3CH$
concentrations above 400 ppt in the measurements, $CH_3CN$ concentrations below 400 ppt was
represented by black dots. During the flight, 16 measurements of acetonitrile concentration above 400
ppt were observed and the maximum was 3227.9 ppt. These observations were located over
northwestern Tennessee and close to the borders of Kentucky, Illinois, Missouri and Arkansas.  Except
for one observation, the flight ASL was between 500 m and 1000 m.
Enhancements of CO and OC were also measured concurrently with $CH_3CN$. Figures 9c and 9d
show scatter plots for $CH_3CN$ versus CO and OC, respectively. Measured $CH_3CN$ was highly correlated to
both measured CO and OC, with linear correlation coefficients ($R^2$) of 0.83 and 0.71, respectively. The
$\Delta CH_3CN/\Delta CO$ ratio is around 2.7 (ppt/ppb) --- consistent with findings of other measurements over
California in 2002 when a strong forest fire signal was intercepted by aircraft (de Gouw et al., 2003). The
$\Delta CH_3CN/\Delta CO$ ratio was around 6.85 (ppt/(mg m$^{-3}$)) ---- in the range of biomass burning analyses in
MILAGRO (Megacity Initiative Local and Global Research Observations) (Aiken et al., 2010).



Figure 9e shows model simulated ΔCO with peaks at AGL below 3000 m. Fire signals showed
substantial influences on aircraft measurement at around 5 UTC. However, clear fire signals between 2
UTC and 3 UTC were observed based on prior CH$_3$CN analysis. The model either predicted insufficient
fire emission influences or missed it. FMS score on July 3 was 30%. Figure 9f shows that CMAQ did not
predict plumes where the HMS plume analysis exhibited several dense smoke plumes. As NOAA Smoke
Text Product (http://www.ssd.noaa.gov/PS/FIRE/DATA/SMOKE) described on its July 03 0501 UTC
report: a smaller very dense patch of remnant smoke, analyzed earlier the same day over southern
Missouri, drifted southward into Arkansas."
The reasons the model missed these fire observations were not clear. Figures 10, 11a and 11b
suggest a few clues. Figure 10 is a backward trajectory analysis plot for the observations obtained during
the SENEX flight on July 3 with CH$_3$CN measured concentration above 400 ppt. Both the transect and
flight altitude of the air parcels clearly showed those measurements were most likely influenced by the
nearby pollution sources. Figure 11a illustrates the locations of fire used in the CMAQ simulation. It is
noted that hmshysplit.txt is input into BlueSky after HMS quality control (Fig. 1). There were several
hotspots around the region where the IMPROVE site MACA was located and where the SENEX flight
overpassed. Our fire simulation system might have underestimated smoke emissions from those fires.
Other explanation was from Fig. 11b, which illustrated hotspots in hmx.txt. In hmx.txt --- showing every
detected fire spots by HMS before quality control. Comparing Fig. 11a with 11b, there were clusters of
fire spots in the central U. S. especially in West Tennessee. However, those spots were removed during
the HMS quality control process because there were no associated smoke plumes visible. In most of
times, those fires were believed to be small sized fires such as from agriculture fires or prescribed burns.
For this case, there seem to have been thin clouds overhead and thicker clouds in the vicinity,
(http://inventory.ssec.wisc.edu/inventory/image.php?sat=GOES-13&date=2013-07-



03&time=16:02&type=Imager&band=1&thefilename=goes13.2013.184.160147.INDX&coverage=CONUS
&count=1&offsettz=0), so it would be hard to differentiate smoke from clouds by satellite observations
**CONCLUSIONS**

In support of the NOAA SENEX field experiment in June-July 2013, simulations were conducted

including smoke emissions from fires. In this study, a system accounting for fire emissions in a chemical
transport model is described, including a satellite fire detecting system (HMS), a fire emission calculation
model (BlueSky), a pre-processing of fire emissions (SMOKE), and simulation over the SENEX domain by
CMAQ. The focus of this work is to evaluate the system's capability to capture fire signals identified by
multiple observation data sets. These data sets included IMPROVE ground station observations, satellite
observations (HMS plume shapefile and ASDTA) and airborne measurements from the SENEX campaign.

For IMPROVE data, potential fire signals were identified by measured potassium concentrations

in $PM_{2.5}$. Fire identifications in CMAQ rely on its predicted $\Delta CO$ and the difference between simulations
with and without fire emissions. Three out of four observed fire signals were captured by CMAQ
simulations. For HMS smoke plume shapefiles that were manually plotted by analysts to represent the
regions impacted by smoke, we used FMS to calculate the percentage of its overlapping with CMAQ
predicted smoke plumes. FMS averaged 22% over forty days of the SENEX campaign.  In terms of fire
smoke impacts on $\Delta AOD$, both ASDTA and CMAQ showed similar patterns that were compared with
HMS plume shapefile analysis. In terms of measured $CH_3CN$, a biomass burning plume tracer, both
SENEX aircraft in-flight measurements and CMAQ simulations captured signatures of long range
transport of fire emissions.

Generally, using HMS-detected fire hotspots and smoke data was useful for predictions of fire

impacts and their evaluation. The HMS-BlueSky-SMOKE-CMAQ fire simulation system, which is also used



in NAQFC, was able to capture most of the fire signals detected by multiple observations. However, the
system failed to identify fire cases on June 17 and July 3 2013 -- thereby demonstrating two problems
with the simulation system. One identified problem was the lack of a dynamical fire LBC outside the
CONUS domain to represent the inflows of strong fire signals originating from outside the simulation
domain. Secondly, the HMS quality control procedure eliminated fire hotspots that were not associated
with visible smoke plumes leading to an underestimation.
We were keen on understanding and quantifying the various uncertainties and observational
constraints of this study therefore the following rules of thumb were observed: (1) A holistic evaluation
approach was adopted so that the fire smoke algorithm was interpreted as a single entity to avoid
impasse arose by uncertainties specific to the different components in the system, (2) Analysis
conclusion applicable to the entire simulation period was drawn so that the episodic characteristics of
the cases embedded in the simulation were averaged and generalized. This new methodology may
benefit NAQFC, (3) We took advantage of the multiple perspectives of the observation systems that
offered a wide spectrum of temporal and spatial variabilities intrinsic to the systems, and (4) We were
intentional to be conservative in discarding data so that we maximized the sampling pool for statistical
analysis and avoided unwittingly discarding poorly simulated cases, good out-layers, and weak sparse
but accurate signals.
Quantitative evaluation of fire emissions and their subsequent influences on ozone and
particulate matter in this fire and smoke prediction system is challenging. Future work includes applying
these findings to the NAQFC and improving the NAQFC system's capabilities to simulate fires accurately.





## Code Availability

The source code used in this study is available online at
http://www.nco.ncep.noaa.gov/pmb/codes/nwprod/cmaq.v5.0.2.

## Acknowledgements & disclaimer

This work was partially funded by the NASA Air Quality Applied Sciences Team (AQAST), project
grant NNH14AX881. The authors are thankful to Dr. Joost  De Gouw and Dr. Martin G. Graus of the Earth
System Research Laboratory, NOAA for sharing the SENEX campaign data used in this study. Although
this work has been reviewed by the Air Resources Laboratory, NOAA and approved for publication it
does not necessarily reflect their policies or views.

## Figures:

Figure 1, Fire emission calculation and smoke plume simulation algorithm.
Figure 2, in 4km SENEX domain, (a): the contribution (%) of CO emission from fires occurred inside the
SENEX domain; (b): the contribution (%) of CO flux flowing into the SENEX domain from its boundary
caused by fires burning outside the SENEX domain but inside the CONUS domain.
Figure 3, simulated ΔCO (ppb) extracted along SENEX flight path.
Figure 4, ΔCO (>2.0 ppb) simulated in SENEX domain on June 24 2013. The solid circle is detected fire
hotspots by HMS. The open triangle represents IMPROVE sites.
Figure 5, FMS (Figure of Merits in Space) (%) from June 11 to July 19 in 2013 during SENXE experiment.
Figure 6, HMS observed plume shape versus CMAQ predicted plume shape on (a): July 6 2013; (b): June
17 2013; The light blue shading represents modeled plume shape (defined as total column ΔCO) and the
thin dash line and emboldened green lines encircle areas representing HMS-derived light and strong
influenced plume shape, respectively. (c): HMS observed fire hotspots (red) and plume shapes (white)
(http://ready.arl.noaa.gov/data/archives/fires/national/arcweb) on June 17, 2013.
Figures 7, GOES detected AOD influenced by fires using ASDTA diagnose method. Color-shaded region
represents the fire-smoke influenced areas and the color denotes the magnitude of the retrieved AOD
on (a): June 14 2013; (d): June 25 2013; ΔAOD (with-fire – without-fire) simulated in CMAQ on (b): June





14 2013; (e): June 25 2013; HMS observed fire hotspots (red) and plume shapes (white) (http://ready.arl.noaa.gov/data/archives/fires/national/arcweb) on (c ): June 14 2013; (f): June 25 2013.

Figure 8, CMAQ simulated ΔCO vertical distributions along SENEX flight transect on (a): June 16 2013; (b): July 10 2013; The x-axis label is UTC (hour) and the y-axis label is AGL (m).Two color bars represent observed $CH_3CN$ concentration (rectangle bar in ppt) and simulated ΔCO concentration (fan bar in ppb), respectively.

Figure 9, plots for July 3 2013 case, (a): IMPROVE; (b): the flight path of SENEX #0703 colored by measured $CH_3CN$ concentration (ppt); (c): $CH_3CN$ (ppt) vs CO (ppb); (d): $CH_3CN$ (ppt) vs AMS_Org (mg m$^{-3}$); (e): CMAQ simulated ΔCO vertical distributions along flight transect; (f): HMS observed plume shape versus CMAQ prediction.

Figure 10, a backward trajectory analysis for the observations obtained during the SENEX flight on July 03 2013 with $CH_3CN$ measured concentration above 400 ppt.

Figure 11, detected fire hotspots on July 03 2013 (a): hmxhysplit.txt; (b): hmx.txt.

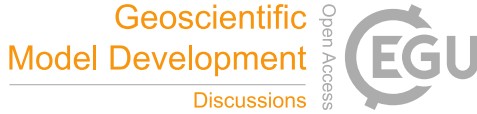



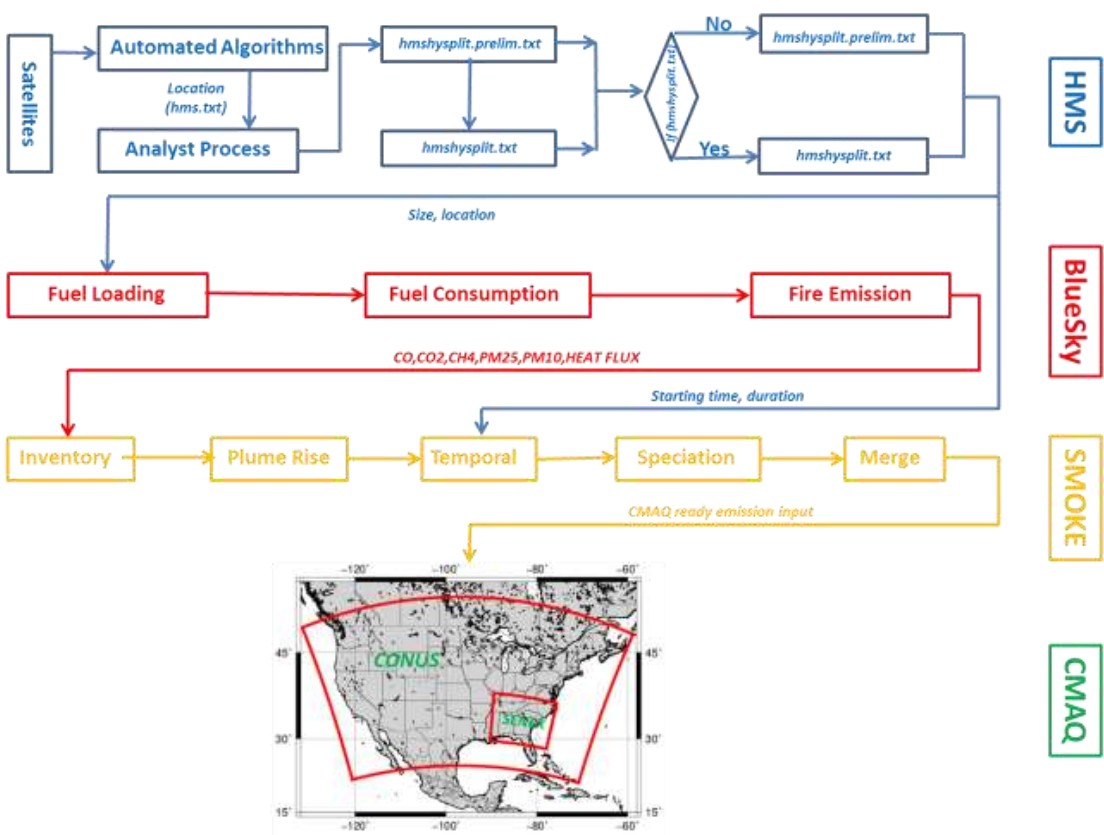


**Figure 1: Fire emission calculation and smoke plume simulation algorithm**






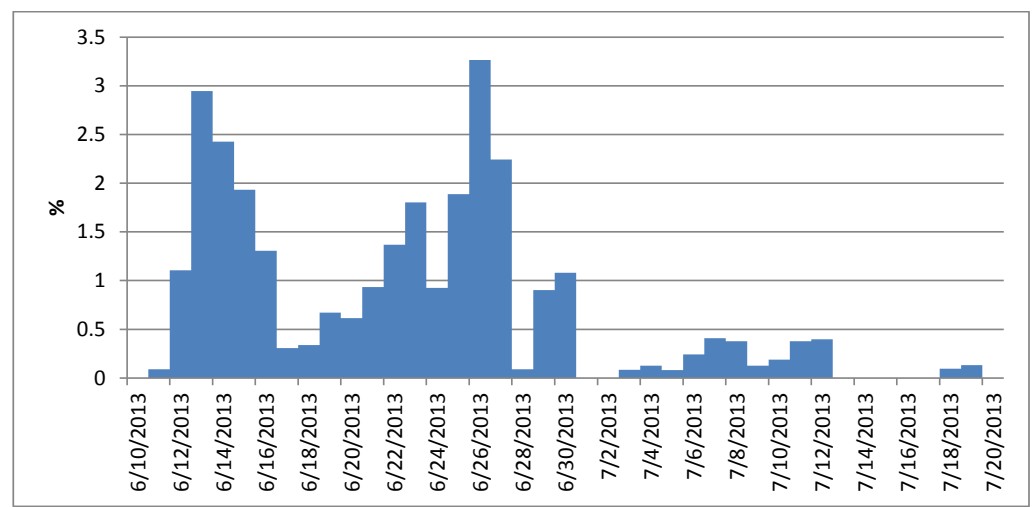


**Figure 2a: the contribution (%) of CO emission from fires occurred inside the SENEX domain**

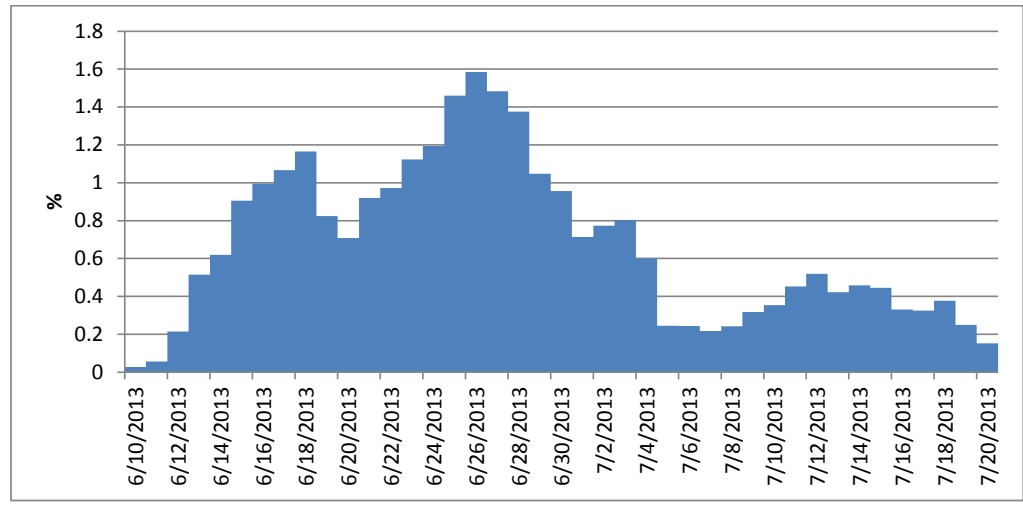


**Figure 2b: the contribution (%) of CO flux flowing into the SENEX domain from its boundary caused by**
**fires burning outside the SENEX domain but inside the CONUS domain**






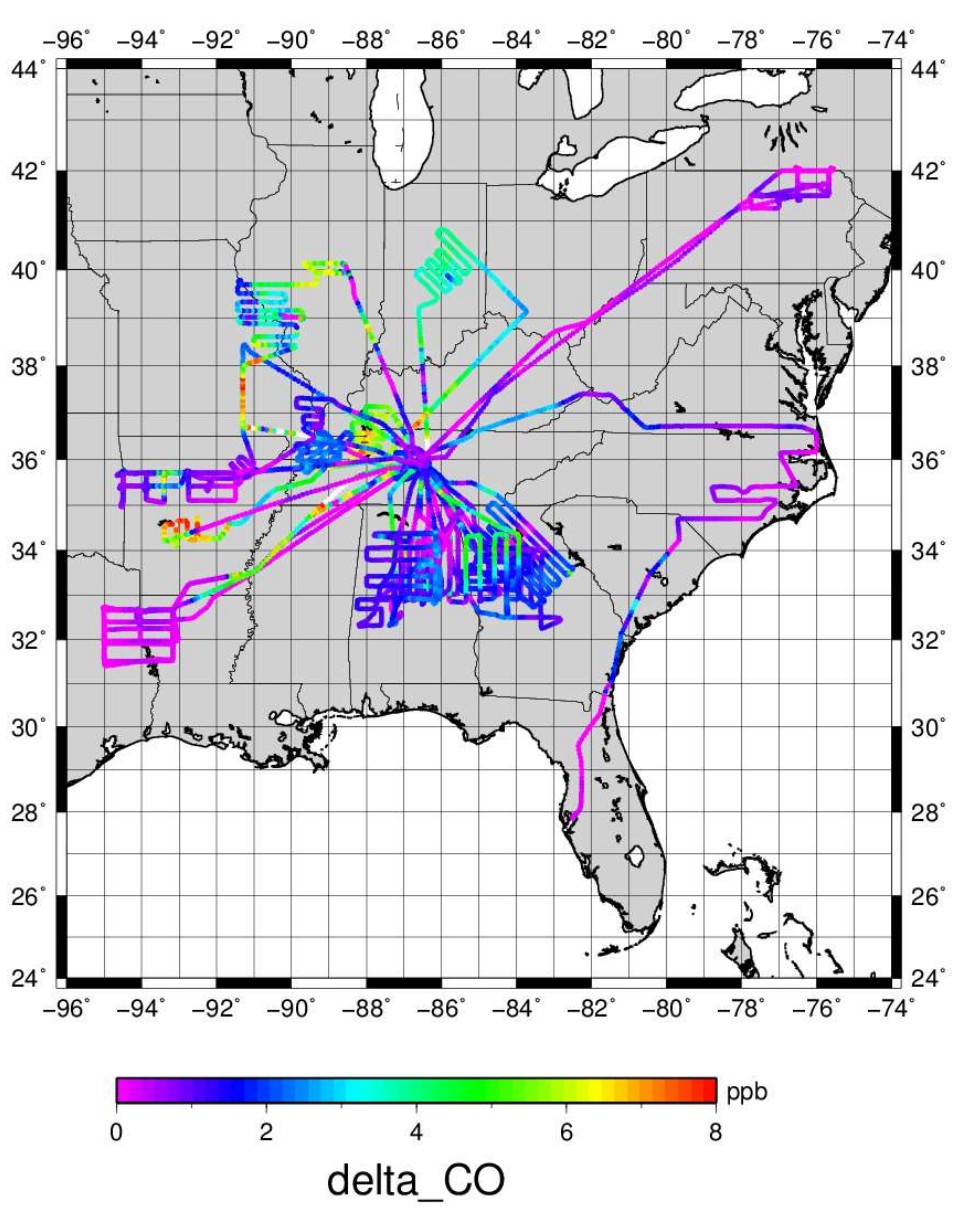


**Figure 3: simulated ΔCO (ppb) extracted along SENEX flight path**






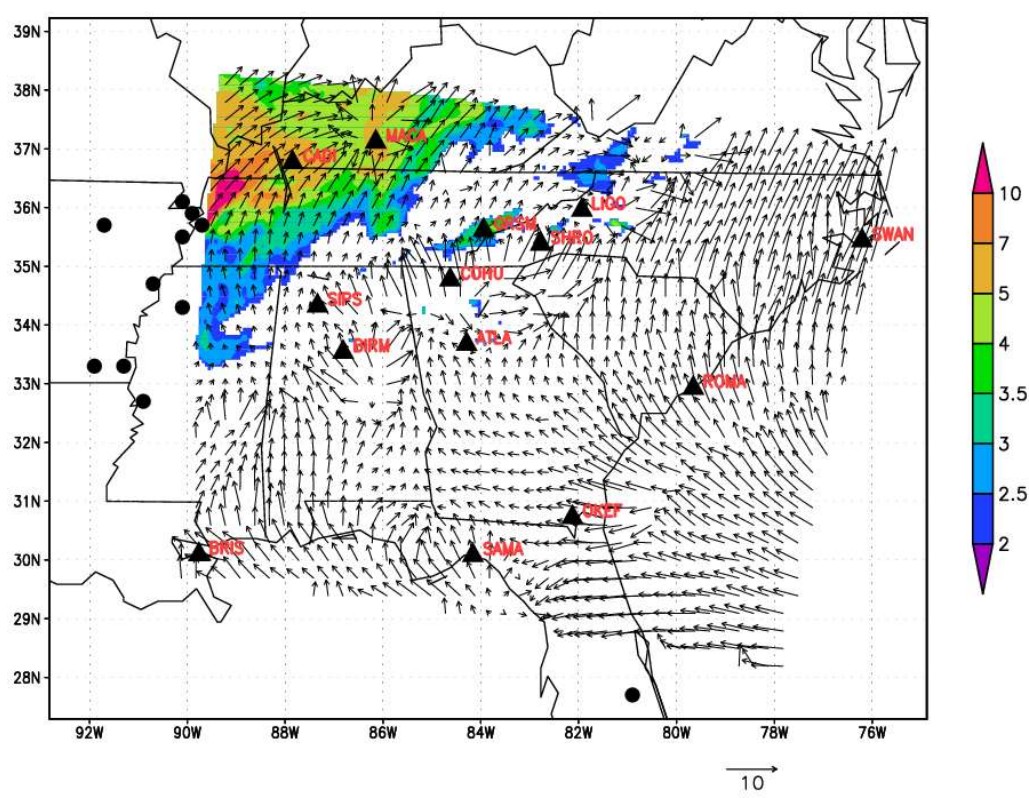


**Figure 4: ΔCO (>2.0 ppb) simulated in SENEX domain on June 24 2013. The solid circle is detected fire hotspots by HMS. The solid triangle represents IMPROVE sites.**













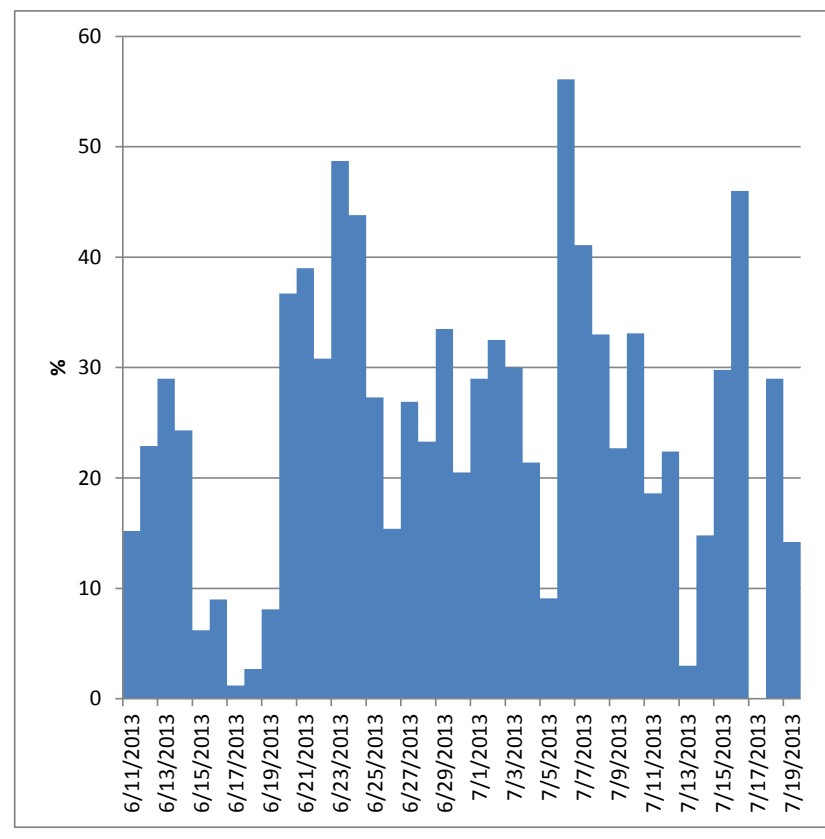


**Figure 5: FMS (Figure of Merits in Space) (%) from June 11 to July 19 in 2013 during SENXE experiment**














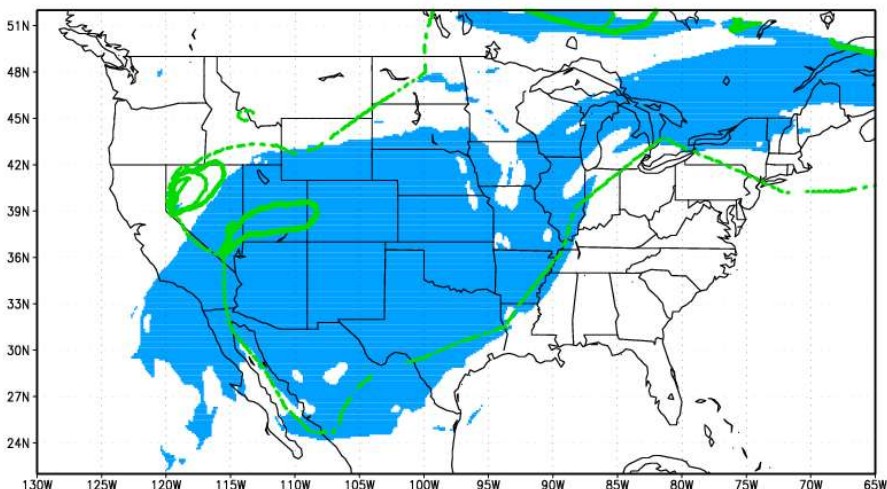


**Figure 6a: HMS observed plume shape versus CMAQ predicted plume shape on July 6 2013; The light blue shading represents modeled plume shape (defined as total column ΔCO) and the thin dash line and emboldened green lines encircle areas representing HMS-derived light and strong influenced plume shape, respectively.**


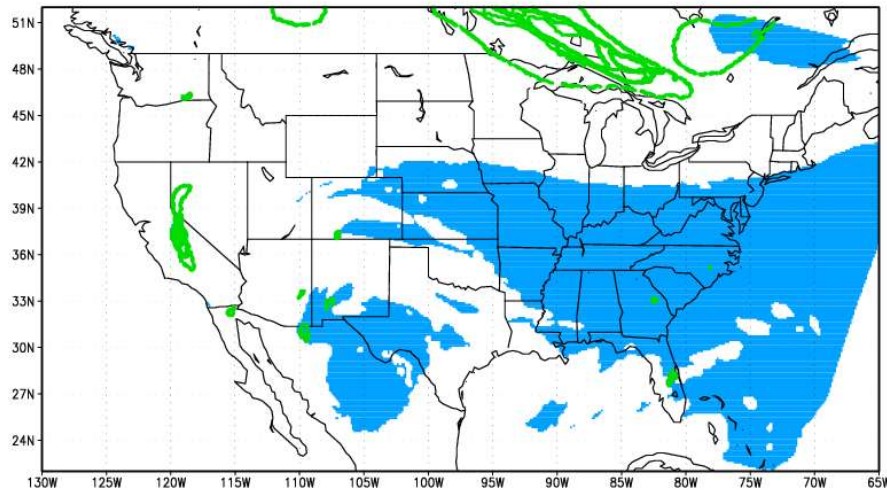


**Figure 6b: on June 17 2013**






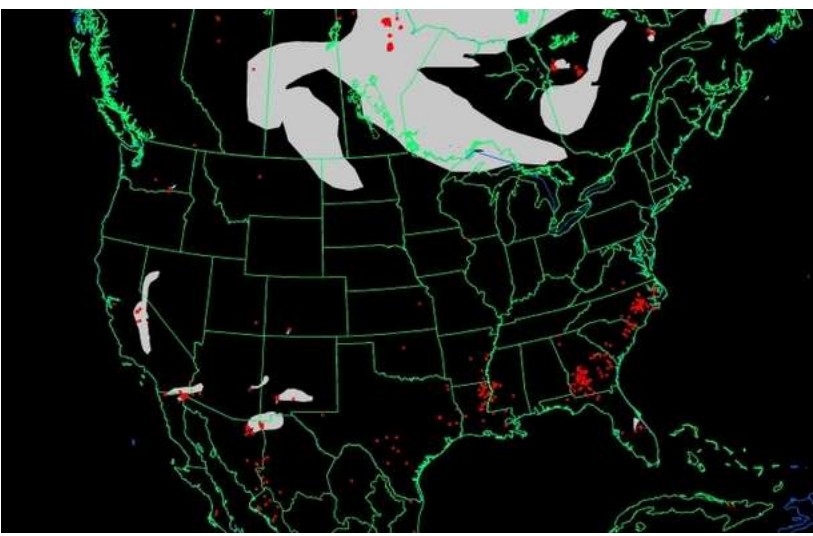


**Figure 6c: HMS detected fire hotspots (red) and smoke plume shapes (white) on June 17 2013**
**(http://ready.arl.noaa.gov/data/archives/fires/national/arcweb)**


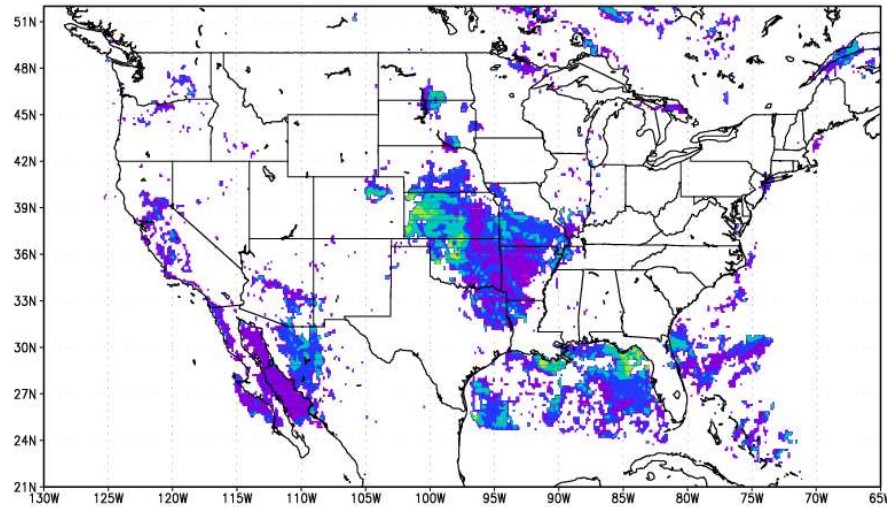


**Figure 7a: GOES detected AOD influenced by fires using ASDTA diagnose method on June 14 2013.**

**Color-shaded region represents the fire-smoke influenced areas and the color denotes the magnitude**
**of the retrieved AOD.**




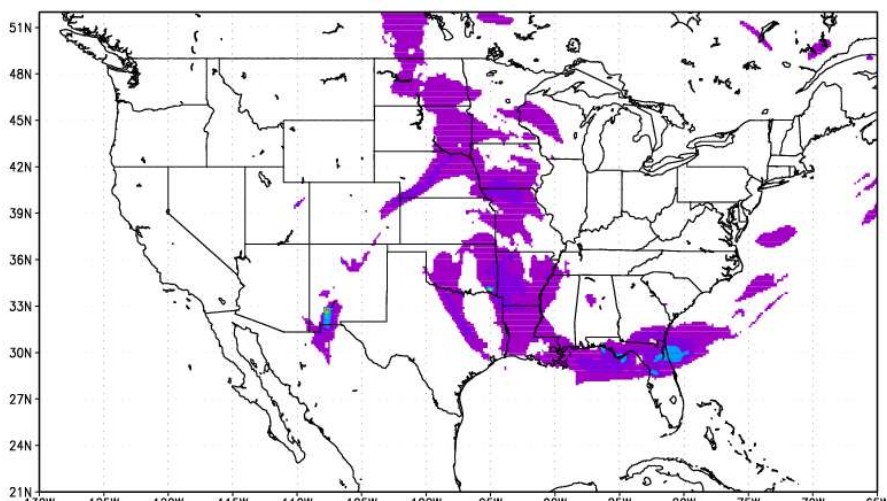


**Figure 7b: simulated ΔAOD (with-fire – without-fire) in CMAQ on June 14 2013**

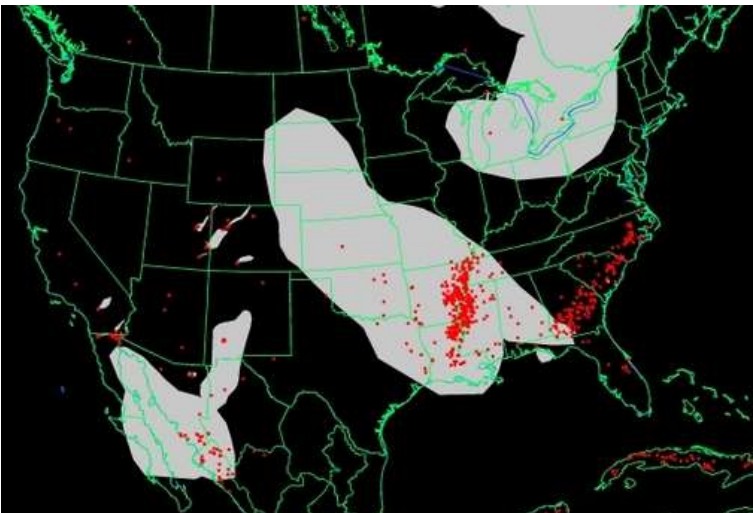


**Figure 7c: HMS detected fire hot spots (red) and smoke plume shapes (white) on June 14 2013**
**(http://ready.arl.noaa.gov/data/archives/fires/national/arcweb)**






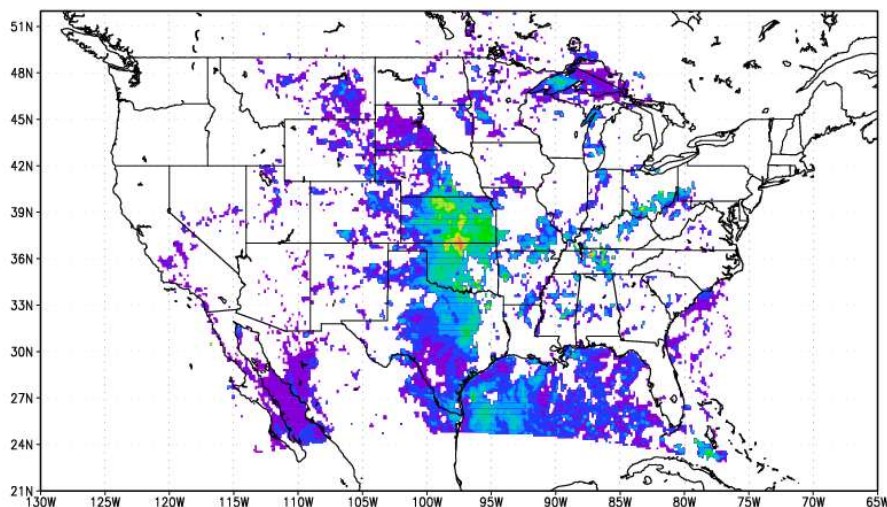


**Figure 7d: GOES detected AOD influenced by fires using ASDTA diagnose method on June 25 2013.**

**Color-shaded region represents the fire-smoke influenced areas and the color denotes the magnitude**

**of the retrieved AOD.**

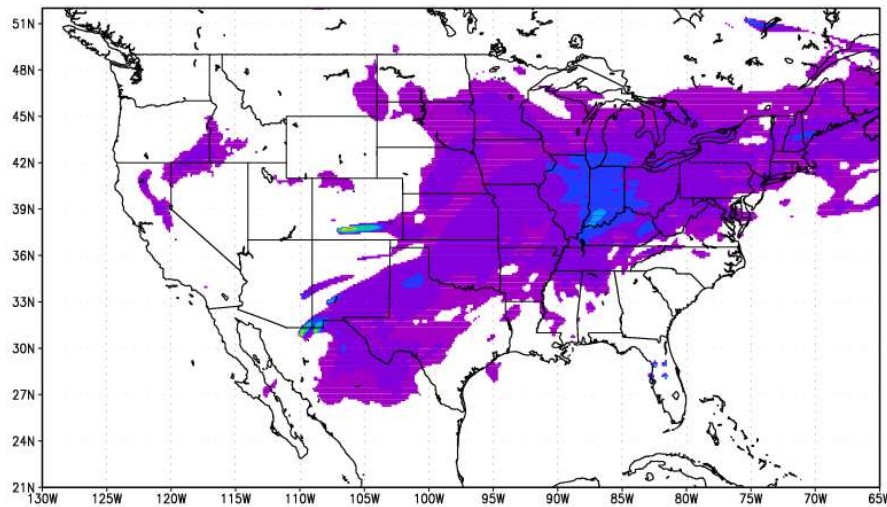


**Figure 7e: simulated ΔAOD (withfire – nofire) in CMAQ on June 25 2013**








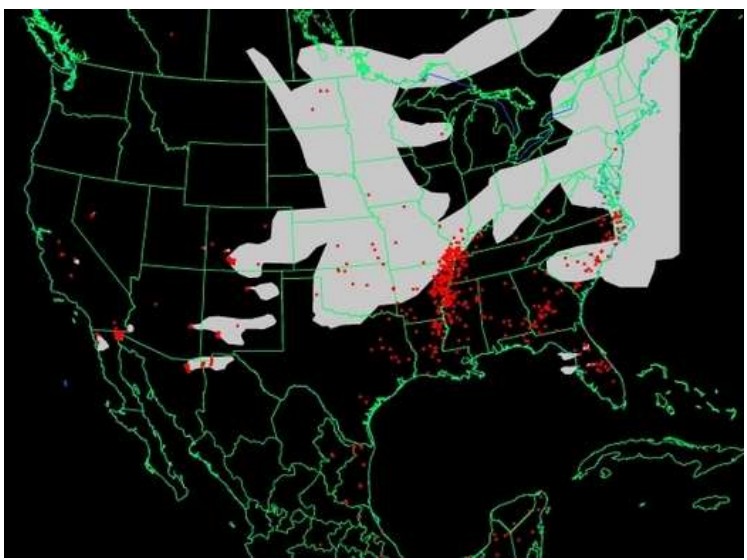

**Figure 7f: HMS detected fire hot spots (red) and smoke plume shapes (white) on June 25 2013**

**(http://ready.arl.noaa.gov/data/archives/fires/national/arcweb)**







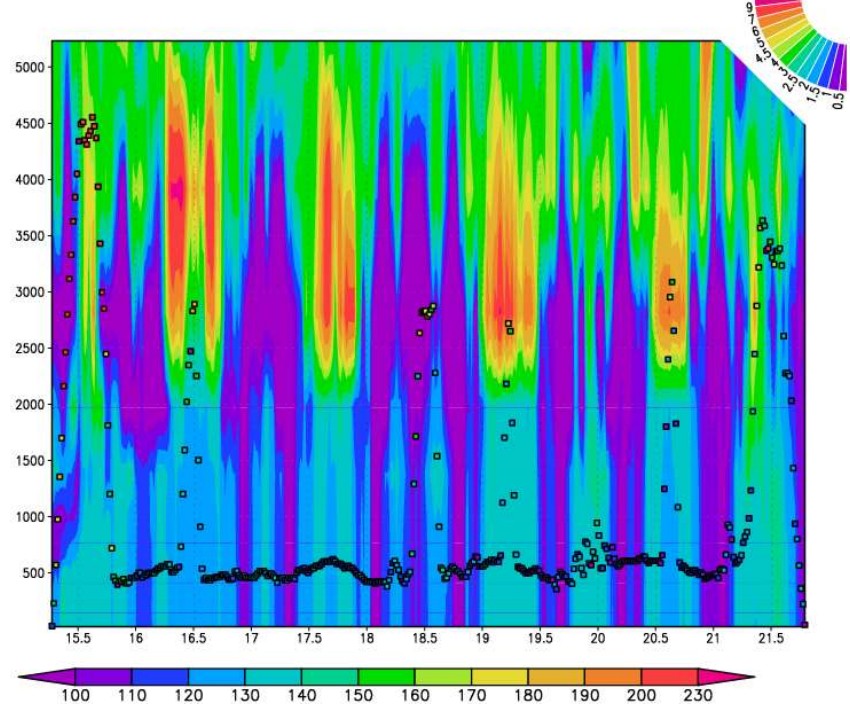


**Figure 8a: CMAQ simulated ΔCO (ppb) vertical distributions along flight transect on June 16 2013. The x-axis label is UTC (hour) and y-axis label is AGL (m). Two color bars represent observed CH₃CN concentration (rectangle bar in ppt) and simulated ΔCO concentration (fan bar in ppb), respectively.**



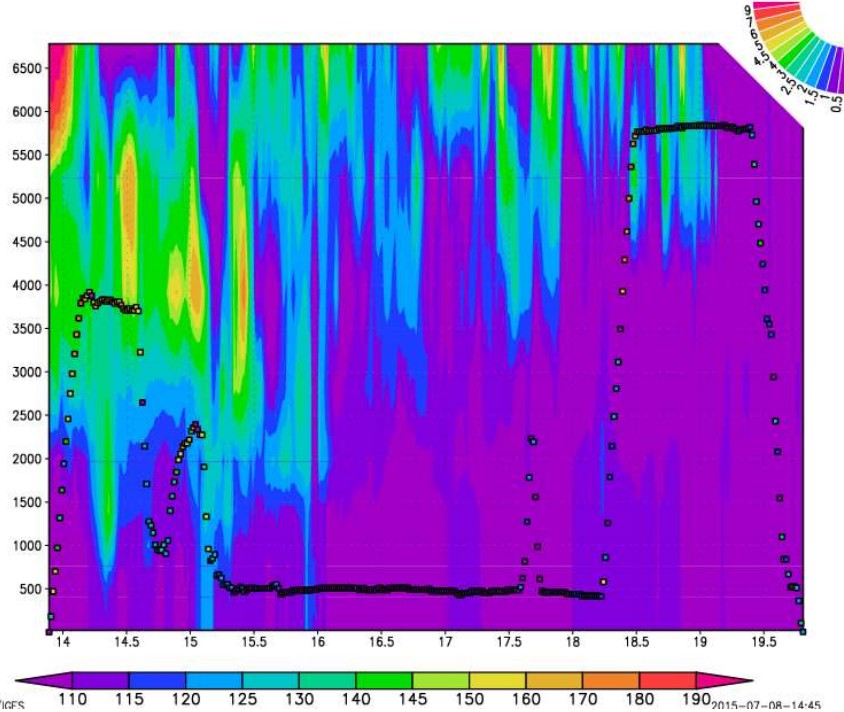



**Figure 8b: CMAQ simulated ΔCO (ppb) vertical distributions along flight transect on July 10 2013. The x-axis label is UTC (hour) and y-axis label is AGL (m). Two color bars represent observed CH$_3$CN concentration (rectangle bar in ppt) and simulated ΔCO concentration (fan bar in ppb), respectively.**








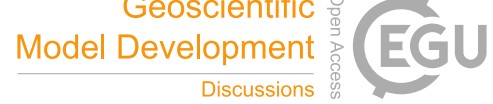



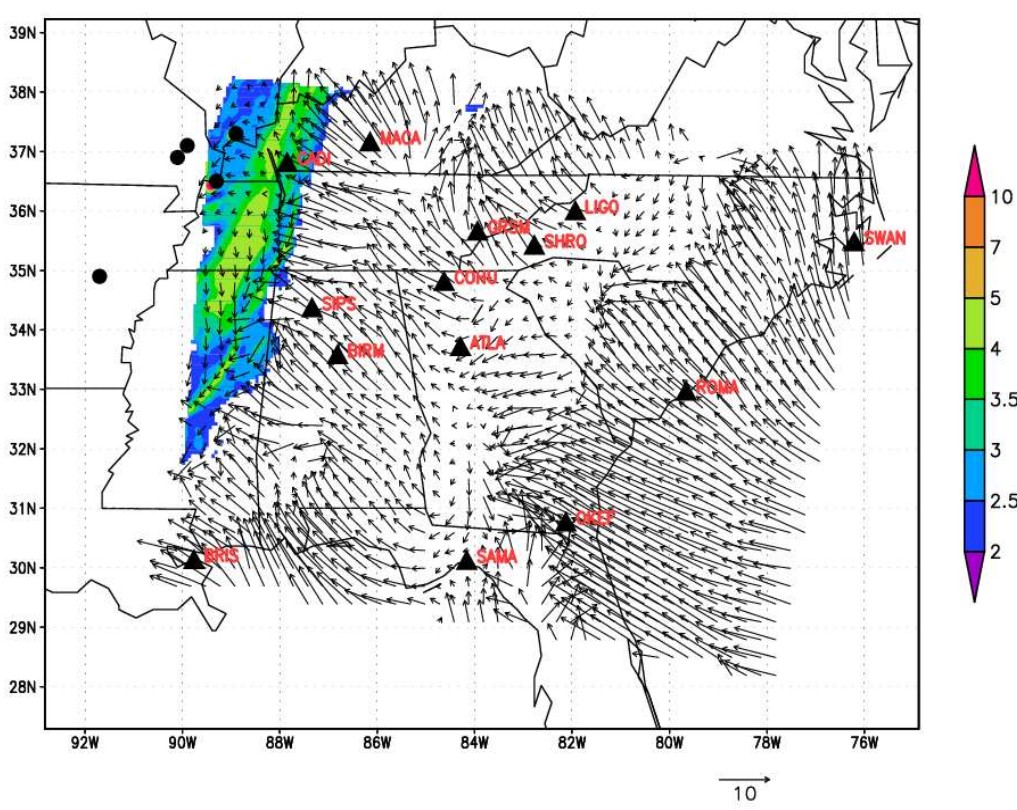


**Figure 9a: ΔCO (>2.0 ppb) simulated in SENEX domain on July 03 2013. The solid circle is detected fire hotspots by HMS. The solid triangle represents IMPROVE sites.**










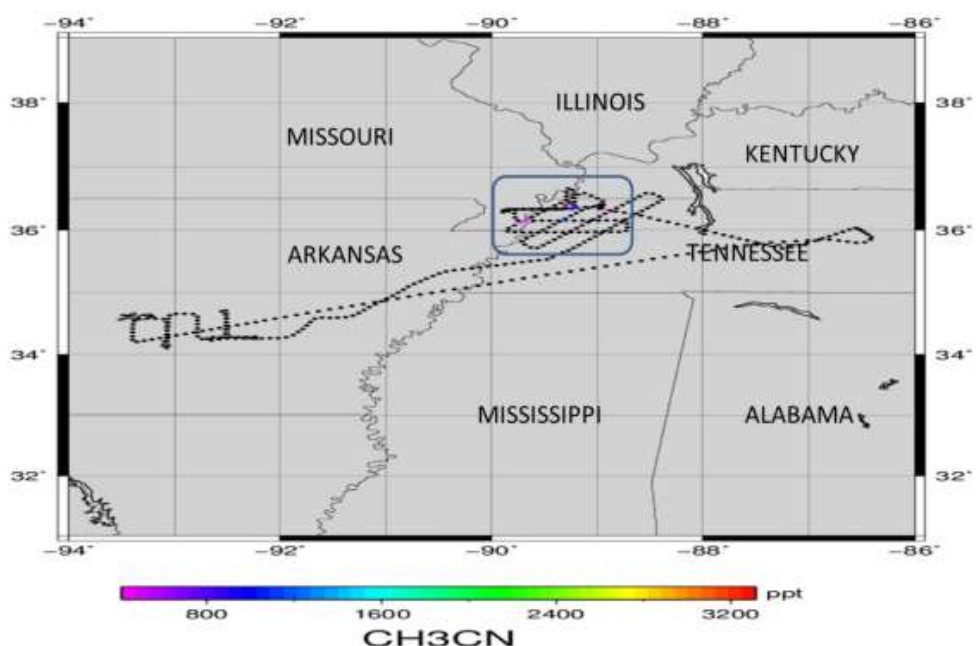


**Figure 9b: the flight path of SENEX #0703, colored by measured CH3CN concentration (ppt)**












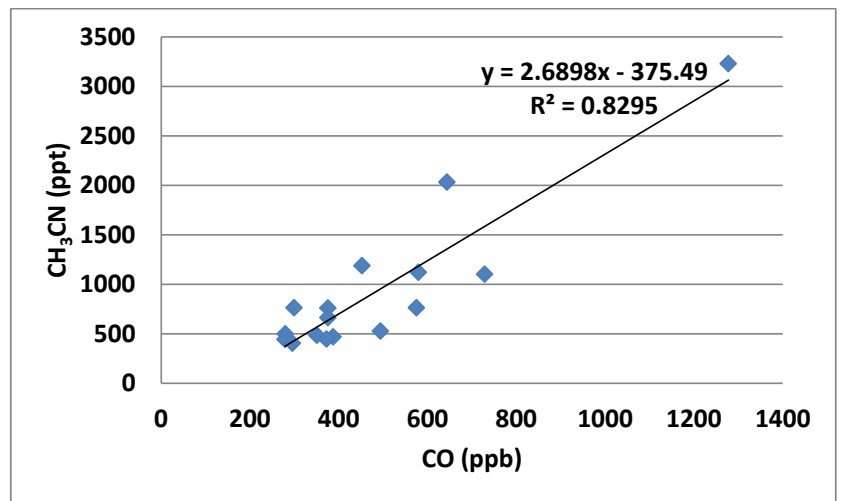


Figure 9c: CH$_3$CN (ppt) vs CO (ppb)

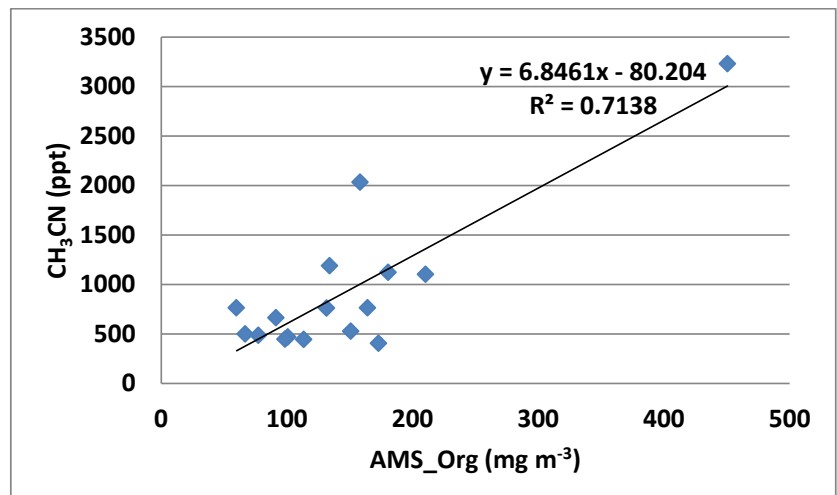


Figure 9d: CH$_3$CN (ppt) vs AMS_Org (mg m$^{-3}$)










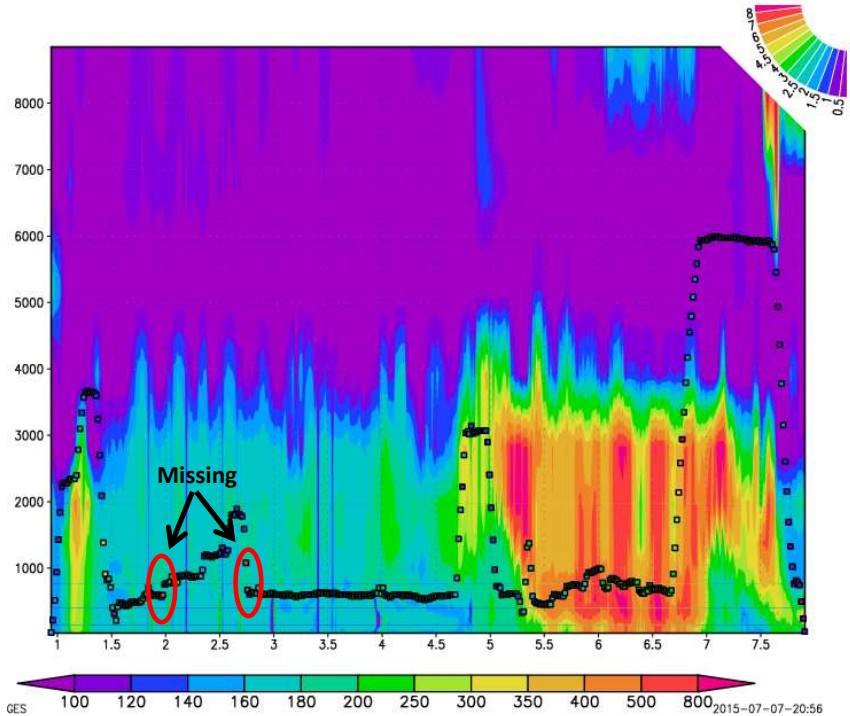


**Figure 9e: CMAQ simulated ΔCO (ppb) vertical distributions along flight transect on July 03 2013. The**
**x-axis label is UTC (hour) and y-axis label is AGL (m). Two color bars represent observed CH₃CN**
**concentration (rectangle bar in ppt) and simulated ΔCO concentration (fan bar in ppb), respectively.**







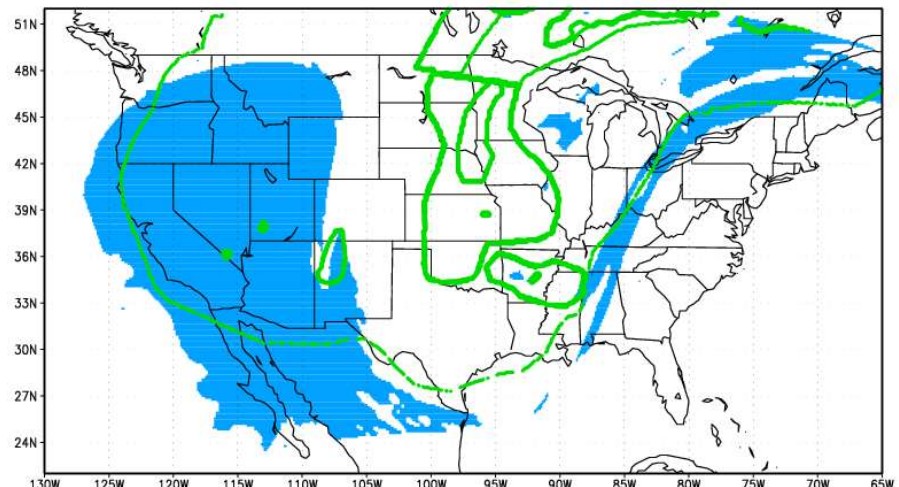


**Figure 9f: HMS plume shape versus CMAQ predictions on July 03 2013. The light blue shading represents modeled plume shape (defined as total column ΔCO) and the thin dash line and emboldened green lines encircle areas representing HMS-derived light and strong influenced plume shape, respectively.**






**Figure 10, a backward trajectory analysis for the observations obtained during the SENEX flight on July 03 with CH$_3$CN measured concentration above 400 ppt.**







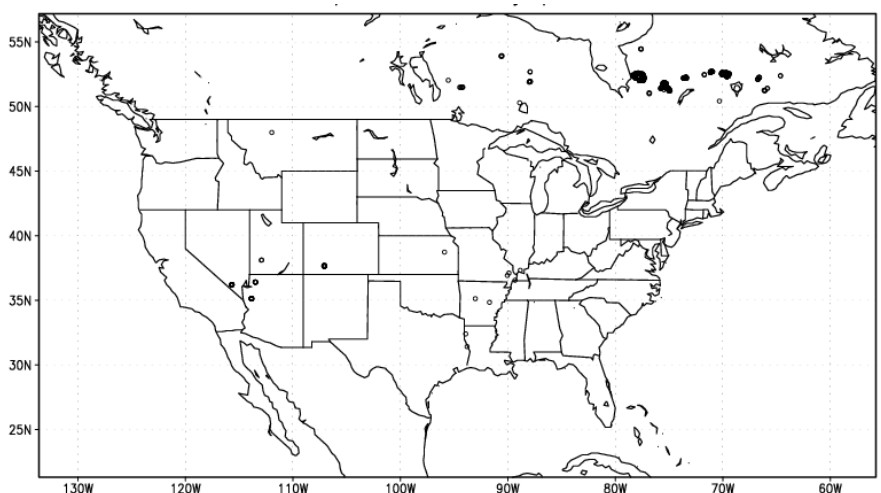


**Figure 11a: fire hotspots in hmxhysplit.txt on July 03 2013**

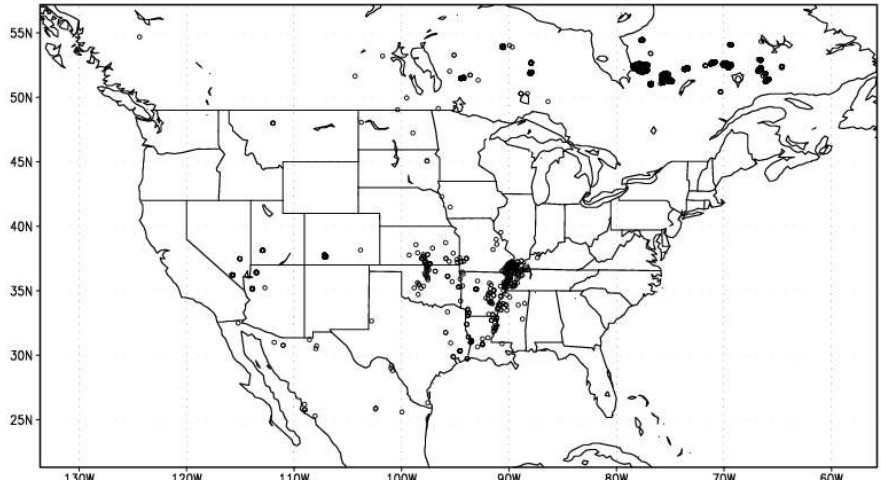


**Figure 11b: fire hotspots in hmx.txt on July 03 2013**





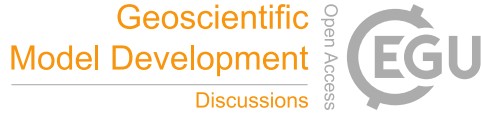



**Tables:**

**Table 1: observed and simulated CO (ppb) during NOAA SENEX experiment**

| AGL (m) | SAMPLE SIZE | OBS | OBS_MAX | Mod_withfire | Mod_nofire | ΔCO |
|---|---|---|---|---|---|---|
| <500 | 166 | 128.93±38.51 | 319.55 | 108.70±21.37 | 107.16±20.34 | 1.54 |
| 500~1000 | 3565 | 146.19±44.39 | 1277.97 | 108.39±19.82 | 106.50±18.86 | 1.88 |
| 1000~1500 | 793 | 125.41±28.09 | 299.64 | 100.11±15.63 | 98.49±14.67 | 1.62 |
| 1500~2000 | 306 | 119.68±23.99 | 265.29 | 100.75±17.04 | 99.08±15.89 | 1.67 |
| 2000~2500 | 219 | 111.48±19.98 | 286.22 | 99.88±17.95 | 98.37±16.92 | 1.51 |
| 2500~3000 | 209 | 111.84±19.79 | 295.79 | 97.43±12.21 | 95.87±11.15 | 1.56 |
| 3000~3500 | 181 | 109.31±16.66 | 197.94 | 89.34±12.09 | 88.13±11.06 | 1.21 |
| 3500~4000 | 195 | 110.78±14.36 | 140.42 | 92.11±10.73 | 90.25±9.62 | 1.86 |
| 4000~5000 | 369 | 89.82±19.09 | 138.04 | 80.36±10.15 | 79.17±9.14 | 1.19 |
| 5000~6000 | 354 | 102.26±22.37 | 209.20 | 78.12±7.64 | 76.82±6.28 | 1.30 |
| 6000~7000 | 85 | 87.53±17.88 | 115.32 | 73.35±4.71 | 70.58±2.45 | 2.77 |

**Table 2: identified fire signals from IMPROVE measurements during SENEX experiment**

| Site | Date | Concentrations (ug m$^{-3}$) | | | | | | Concentration/Average | | | | | | Ratio | |
|---|---|---|---|---|---|---|---|---|---|---|---|---|---|---|---|
| | | EC | OC | K | SOIL | NO$_3^-$ | SO$_4^{2-}$ | EC | OC | K | SOIL | NO$_3^-$ | SO$_4^{2-}$ | BC/OC | K/BC |
| COHU | 0621 | 0.28 | 2.10 | 0.05 | 0.22 | 0.13 | 2.61 | 1.4 | 1.46 | 1.42 | 0.39 | 0.84 | 1.28 | 0.1331 | 0.1933 |
| MACA | 0624 | 0.45 | 2.34 | 0.09 | 0.26 | 0.24 | 2.76 | 1.85 | 1.58 | 1.82 | 0.48 | 1.19 | 1.24 | 0.1929 | 0.1973 |
| MACA | 0703 | 0.33 | 2.32 | 0.08 | 0.16 | 0.29 | 2.11 | 1.35 | 1.57 | 1.73 | 0.29 | 1.43 | 0.94 | 0.1423 | 0.2554 |
| BRIS | 0703 | 0.24 | 0.98 | 0.21 | 0.31 | 0.11 | 2.63 | 1.49 | 1.28 | 2.79 | 0.13 | 0.35 | 1.36 | 0.2458 | 0.8851 |
| GRSM | 0621 | 0.25 | 1.56 | 0.05 | 0.24 | 0.13 | 2.52 | 1.36 | 1.45 | 1.24 | 0.49 | 0.99 | 1.42 | 0.1596 | 0.1979 |



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
