# Peer review of "Evaluating a fire smoke simulation"

_Geoscientific Model Development, 2018_

## Short Comment (SC1) · 18 Dec 2018

Dear authors,

in my role as Executive editor of GMD, I would like to bring to your attention our Editorial version 1.1: http://www.geosci-model-dev.net/8/3487/2015/gmd-8-3487-2015.html This highlights some requirements of papers published in GMD, which is also available on the GMD website in the 'Manuscript Types' section: http://www.geoscientific-model-
development.net/submission/manuscript_types.html In particular, please note that for your paper, the following requirement has not been met in the Discussions paper:

- "The main paper must give the model name and version number (or other unique identifier) in the title."

Please provide a version number of NAQFC in the title of your revised manuscript. Note, that a name and a version number are important to identify your specific developments.

As explained in https://www.geoscientific-model-development.net/about/manuscript_types.html. GMD is encouraging authors to upload the program code of models (including relevant data sets) as supplement or make the code and data of the exact model version described in the paper accessible through a DOI (digital object identifier). In case your institution does not provide the possibility to make electronic data accessible through a DOI you may consider other providers (eg. zenodo.org of CERN) to create a DOI. Please note that in the code availability section you can still point the reader to how to obtain the newest version.

Yours,

Astrid Kerkweg

---

## Short Comment (SC2) · 18 Dec 2018

The naqfc version is cmaq.v5.0.3

---

## Referee Comment (RC1) · Anonymous Referee #1 · 19 Jan 2019

Journal: GMD Title: Evaluating a fire smoke simulation algorithm in the National Air Quality Forecast Capability (NAQFC) by using multiple observation data sets during the Southeast Nexus (SENEX) field campaign Author(s): Li Pan et al. MS No.: gmd-2017-207 MS Type: Model evaluation paper

general comments

The authors utilize a variety of physical & chemical data to evaluate their fire emissions

and air quality modeling system, with a particular effort to utilize the SENEX campaign observations. Their motivation is apparently to provide guidance on use or further development of the very similar NAQFC system, as cited in lines 44-45 in the introduction: '...National Air Quality Forecasting Capability (NAQFC) daily PM2.5 operational forecast (Lee et al., 2017).'

The authors describe numerous analyses they completed to compare the 'fire signals' to be found in the CMAQ model results: deltaCO from CMAQ, PM2.5 CO and EC, acetonitrile, AOD, satellite fire hotspot detects and plume extents.

Since the NAQFC is explicitly cited as being for the purpose of predicting PM2.5, and since this paper seeks to evaluate a NAQFC analogue, it seems quite odd that there was no effort to compare the SENEX EC and OC PM2.5 with model results, except in terms of ratios. While the paper shows considerable and diverse efforts to utilize appropriate data to evaluate the simulation results, poor writing obscures the value and meaning of this work to an unacceptable extent. The paper is authored by a respectable set of scientists; it is hard to believe that most of these authors actually read the paper as reviewed, so rife was it with grammatical errors, confusing word choices, contorted syntax.

specific comments

I saw no effort to directly compare CMAQ PM2.5 with SENEX PM2.5 EC and OC.

Figures 7a, b, d and e are all missing color bar legends for AOD.

In Figures 8a and b and 9e, the square symbols for observations are so densely packed that their outlines (in black) obscure the symbol colors over much of the flight path. Perhaps the density of observation points could be reduced in some areas and/or the symbols made larger to address this.

technical corrections Before submission for review, it is hoped that all the authors of the paper have read and edited the MS. I very much doubt that was the true here, as this

MS is rife with grammatical errors and confusing constructions. I could not offer this to students as a model of scientific writing. Some of these are indicated below, but really, there are so many that I didn't get to all of them.

Line 29, comma after campaign Line 33 change 'helped identified' to showed or identified Line 36 , change 'filter out' to retrieve or to 'focus on' Line 77 change 'comprised' to consists to make this a sentence. Change Satellite to Satellites Line 82 and following: itemization of the file names could maybe be best isolated in supplemental material. It isn't clear that using file names in this discussion adds much. Line 94: is hmx.txt meant to read hms.txt as used above. Line 97: HMS imagery is Line 109 and elsewhere: In remote sensing a 12-km grid does not reliably 'resolve' features of size of 12 km, so I object to this casual misuse of 'resolution', even though it is common. Say '12- km grid' or otherwise describe. Use 12-km as adjective for grid. Be consistent. Line 121-125: Confusing Line 128: emission rates Line 132: gridded emission Line 143: If that's a crude estimate what is a better approach and why wasn't that tested? Line 166-168: confusing. Reword. Line 171: is emitted by biomass burning Line 181-183 Not sure, but this sounds like you intend to tell us which processes contribute how much error. Line 188-190: But apparently not... Line 190: the purpose is to focus on fire/smoke signal timing? Line 205: Table 1 only gives AGL, not ASL. Lines 210 and 214: change exhibits to shows Line 219 – 222: Unclear Line 222-225: Not a sentence, even. Line 227: 'not negligible perspective' is unclear Line 246: change 'below' to 'above' and change 'was' to 'were' Line 252: Change Tab. to Table Line 269: Change 'for' to 'to' Line 284 – 292: Could you say something clarifying about the significance of interference from clouds in making informative FMS comparisons? Line 290: CMAQ didn't underestimate it, the HMS BlueSky SMOKE emissions system did. Line 294-5. No, your system used a climatological LBC and was thus blind to whether there was more or less actual influence from external fires. Line 303: 'a similar analysis' or 'similar analyses'. Line 302: 'is accessed' Line 309: 'Other reasons... are discussed...' Line 331: change sparingly to occasionally or rarely. Line 334: change that to those or change that to were. Line 338: change 'are subject' to tend Line 387: So CH3CN decreased

along with AGL, as AGL decreased? Or was inversely related to AGL? An ambiguously statement as written. Line 398: change 'was' to 'were' Line 403-5: The single isolated CH3CN value of 3000+ strongly affects the slopes in Figures 9c and 9d. Line 444: 'rely on predicted delta CO, the difference....' Line 449: delete 'similar', change 'compared with' to 'comparable to' Line 450: change 'shapefile analysis' to 'shapefiles' Line 452: end sentence as 'from elsewhere in the CONUS domain.' Line 457: change 'outside' to 'bounding' Line 461 – 471: For a structure like this with a colon leading to a list of independent clauses (1-4) (that may or may not contain commas), begin each clause in lower case and terminate all the clauses with a semicolon. Except for the last one, which gets a period. Line 463-4: 'to avoid impasse arose by uncertainties' is unclear 468-9: we were intentionally conservative... Line 470: 'outliers' and delete 'sparse' Format used in text for citation is inconsistent.

---

## Referee Comment (RC2) · Anonymous Referee #2 · 2 Aug 2019

This article conducts the evaluation the NAQFC simulations including fire smoke particulate (PM25) emission using observations from in-situ, aircraft, and satellite measurements. Several useful indicators/methodologies had been described in this article to identify the signal of fire smoke influence. This article shows valuable information on future evaluation of the impact of fire smoke emission on modeled PM25, as well as the improvement of air quality modeling. However, the manuscript may need major revision to polish its statements for reader to easily understand the message that authors want to deliver. I often found myself taking too much time trying to understand what authors want to say in a paragraph and between paragraphs. This is a common problem of the writing of this manuscript. It lacks transition wording to connect idea between sentences in a paragraph as well as between paragraphs, e.g., the paragraph [lines 461-471] discussed below. I encourage lead author to work closely with co-authors to make the reading easier to deliver the value of this study.

General comments

(1) It may be just a personal preference issue, but I suggest authors to rewrite sentence started with "we will compare…." or "our simulation…" TO "this study will…", "the results show…", "the comparison between A and B indicates…".

(2) Replace current sentence using "- -" with a complete sentence, e.g., lines 339 and 408.

(3) Some description belong to figure or table caption and can be removed from main body. It may be easier to understand the main issue, e.g., Lines 323 to 328.

(4) Avoid adding a single (maybe unrelated) sentence in the middle of a paragraph to stop the flow of message, e.g., line 317 "The ASDTA is a signature identification analysis.". Do not try to clog the article with extra information. Just a few simple and focused descriptions can better deliver your message.

Specific comments:

(1) Lines 76-79:

a. The composition of HMS sources are different now from the time this manuscript submitted. To avoid confusion, please add "At the time of this study" at the beginning of the paragraph.

b. MODIS and AVHRR is sensors while GOES-12, NASA EOS Aqua, and NOAA-15 …etc. are satellites. Please spell out 15/17/18 as NOAA-##. Consider using [.…..the fire detection from "sensor" on-board "satellite".……].

(2) Lines 240-249:

a. How did authors come up with threshold values, i.e., > 20%, < 50%, and < 1? Please provide the reference of the source of the threshold.

b. Please add "ratio" to the column title of table 2, for columns 9-14.

c. My understanding of this paragraph is the ratio should be > 1.2 for EC, OC, and K, < 0.5 for NO3- and SO42-, and < 1 for soil to be classified as "influence by fire smoke". But Table 2 shows NO3- and SO42- ratios at COHU, MACA (two date), and GRSM do not satisfy the criterion, is my understanding wrong? Maybe simply spelling of conditions based on ratio values, such as ratio A > thrershold 1, ratio B < threshold 2, and ratio C >= threshold 3.

(3) Lines 312-315

a. My knowledge about ASDTA indicates the description of ASDTA is incorrect. AS-DTA uses satellite observed AOD and meteorological fields from the NCEP operational meteorology model. It does not use HYSPLIT model simulated output. Authors should verify their description with NOAA NESDIS developers of ASDTA.

b. If (a) is correct, please replace all "predicted" ASDTA products with "diagnosis" ASDTA products in manuscript.

(4) Lines 341-348 are difficult to understand. My guessing is the authors trying to explain why CMAQ can not capture the fire signal because of (a) do not have a dynamic LBC including the trans-boundary influence of fire smoke PM25 originated from fires outside modeling domain (b) plume rise scheme difference, and (c) different number of fire hotspot used. (c) May not be totally correct, in my opinion, the number of hotspot difference is attributed to difference of domain coverage where HYSPLIT domain is larger. The different model performance between CMAQ and HYSPLIT is already explained by (a), i.e., the HYSPLIT can simulate the long rang transport impact of Canadian fires because it has the fires within its domain.

(5) Line 399, the first appearance of "acetonitrile" in this manuscript. Is it CH3CN? Otherwise there is no description in previous paragraphs that this chemical species can be used to identify fire signal.

(6) Lines 461-471

This paragraph show-up from nowhere and it seems to me has no connection to this study. It is more like a personal experience on the difficulty of fire smoke modeling. I do not know whether items 1-4 are concluded as a result from diagnoses of this study, from a common knowledge of the community, or simply speculation?

Since I really have trouble to comprehend the paragraph, I am going to make a bold guess and recommend authors to re-word this paragraph as

The comparison of A in this study shows [item 1]. But [item 2] of this study indicates there are other factors. It is commonly known that [item 3] can impact the results. Thus [item 4] found this study can be used to improve [item 5]. ...etc.

(7) Color bar is needed for Figures 7a, 7b, 7d, and 7e, otherwise simple description is needed to let reader know the direction of changing color corresponds to the increase/decrease. Also, those figures are colored-shaded plot. They are not contour plot. The description of figures should be corrected in manuscript.

(8) Figures 9b. Can not see the color of circles for CH3CN concentration.

―――――――――――――――――――

---

## Author Comment (AC1) · 31 Aug 2019

General comments "The authors utilize a variety of physical & chemical data to evaluate their fire emissions and air quality modeling system, with a particular effort to utilize the SENEX campaign observations. Their motivation is apparently to provide guidance

on use or further development of the very similar NAQFC system, as cited in lines 44-45 in the introduction:': : :National Air Quality Forecasting Capability (NAQFC) daily PM2.5 operational forecast (Lee et al., 2017).' The authors describe numerous analyses they completed to compare the 'fire signals' to be found in the CMAQ model results: deltaCO from CMAQ, PM2.5 CO and EC, acetonitrile, AOD, satellite fire hotspot detects and plume extents. Since the NAQFC is explicitly cited as being for the purpose of predicting PM2.5, and since this paper seeks to evaluate a NAQFC analogue, it seems quite odd that there was no effort to compare the SENEX EC and OC PM2.5 with model results, except in terms of ratios. While the paper shows considerable and diverse efforts to utilize appropriate data to evaluate the simulation results, poor writing obscures the value and meaning of this work to an unacceptable extent. The paper is authored by a respectable set of scientists; it is hard to believe that most of these authors actually read the paper as reviewed, so rife was it with grammatical errors, confusing word choices, contorted syntax."

Response: First of all, we'd like to thank reviewer efforts in reviewing this manuscript and valuable comments. For two major concerns raised by reviewer, we will address them in the following response.

Specific comments "I saw no effort to directly compare CMAQ PM2.5 with SENEX PM2.5 EC and OC."

Response: The main focus of this manuscript was to evaluate fire smoke algorithm used in NAQFC. The SENEX campaign observed PM2.5 concentration was one of the data sets this study used in validation. The philosophy being that NAQFC pays more attention to surface PM2.5 as it afflicts human health significantly (Brauer et al., 2015). We customarily use surface PM2.5 observation instead of flight measurement PM2.5 in NAQFC evaluation (Chai et al., 2013; Lee et al., 2017; Pan et al., 2014).

"Figures 7a, b, d and e are all missing color bar legends for AOD." Response: Graphs have been redrawn.
"In Figures 8a and b and 9e, the square symbols for observations are so densely packed that their outlines (in black) obscure the symbol colors over much of the flight path. Perhaps the density of observation points could be reduced in some areas and/or the symbols made larger to address this."

Response: Figures 8a, 8b and 9e have been modified.

"Line 29, comma after campaign" Response: It has been modified.

"Line 33 change 'helped identified' to showed or identified" Response: It has been modified.

"Line 36 , change 'filter out' to retrieve or to 'focus on' " Response: It has been changed to retrieve.

"Line 77 change 'comprised' to consists to make this a sentence. Change Satellite to Satellites" Response: It has been modified.

"Line 82 and following: itemization of the file names could maybe be best isolated in supplemental material. It isn't clear that using file names in this discussion adds much." Response: Figure 1 uses these file names, which represents the order of steps in the process of HMS. Simulation results are significantly affected by the files used in model, for example, in SENEX case #0703.

" Line 94: is hmx.txt meant to read hms.txt as used above. " Response: No, it means that detected wildfire information included in hmx.txt.

"Line 97: HMS imagery is " Response: It has been modified.

"Line 109 and elsewhere: In remote sensing a 12-km grid does not reliably 'resolve' features of size of 12 km, so I object to this casual misuse of 'resolution', even though it is common. Say '12- km grid' or otherwise describe. Use 12-km as adjective for grid. Be consistent. " Response: It has been changed to"12-km CMAQ model grid".

"Line 121-125:Confusing" Response: The actual situation is such.

"Line 128: emission rates " Response: It has been modified.

"Line 132: gridded emission" Response: It has been modified.

"Line 143: If that's a crude estimate what is a better approach and why wasn't that tested?" Response: HMS doesn't provide such information. Constant profile is the assumption at the time.

"Line 166-168: confusing. Reword. " Response: Rewrote it into "The analytical run is a 24-hour retrospective simulation using yesterday's meteorology and fire emissions to provide initial conditions for today's forecast. The forecasting run is a 48-hour predictive simulation using yesterday's fire emissions, assuming fires with duration of more than 24 hours are projected as continued fires."

"Line 171: is emitted by biomass burning " Response: It has been modified.

"Line 181-183 Not sure, but this sounds like you intend to tell us which processes contribute how much error." Response: Rewrote it into "In this study, we realized that it is almost impossible to assess the uncertainty of each specific smoke physical process"

"Line 188-190: But apparently not..." Response: It has been modified.

"Line 190: the purpose is to focus on fire/smoke signal timing? " Response: It has been modified "to capture fire signals".

"Line 205: Table 1 only gives AGL, not ASL. " Response: Table 1 shows CMAQ simulated results, which is based on AGL.

"Lines 210 and 214: change exhibits to shows" Response: It has been modified.

"Line 219 – 222: Unclear" Response: This means above average $\Delta$CO concentration.

"Line 222-225: Not a sentence, even. " Response: It has been modified to "For an example, a clear fire signal between 500 m and 1000 m AGL was indicated by $\Delta$CO across those altitudes and when the concentration of $\Delta$CO was above 2.0 ppb – based

on the campaign duration averaged CO concentrations of about 150 ppb as well as on within the SENEX domain and outside of SENEX domain fire contributions to CO (150*(0.007+0.0067) =2.0)."

"Line 227: 'not negligible perspective' is unclear" Response: It has been modified.

"Line 246: change 'below' to 'above' and change 'was' to 'were' " Response: It has been modified.

"Line 252: Change Tab. to Table" Response: It has been modified.

"Line 269: Change 'for' to 'to'" Response: It has been modified.

"Line 284 – 292: Could you say something clarifying about the significance of interference from clouds in making informative FMS comparisons? " Response: June 17th case as an example was discussed in line 293-300.

"Line 290: CMAQ didn't underestimate it, the HMS BlueSky SMOKE emissions system did. " Response: It has been changed to "HMS-BlueSky-Smoke emission system".

"Line 294-5. No,your system used a climatological LBC and was thus blind to whether there was more or less actual influence from external fires. " Response: At that time, NAQFC used climatological LBC. Now, dynamic boundary condition from NGAC is used in NAQFC (Wang et al., 2018).

"Line 303: 'a similar analysis' or 'similar analyses'." Response: It has been modified.

"Line 302: 'is accessed' Response: It has been modified.

Line 309: 'Other reasons: : : are discussed: : :'" Response: It has been modified.

" Line 331: change sparingly to occasionally or rarely." Response: It has been modified.

"Line 334: change that to those or change that to were." Response: It has been modified.

"Line 338: change 'are subject' to tend" Response: It has been modified.

"Line 387: So CH3CN decreased along with AGL, as AGL decreased? Or was inversely related to AGL? An ambiguously statement as written." Response: "the decrease with AGL" has been deleted.

"Line 398: change 'was' to 'were'" Response: It has been modified.

" Line 403-5: The single isolated CH3CN value of 3000+ strongly affects the slopes in Figures 9c and 9d." Response: The enhancements of CO and OC were also measured at same moment.

"Line 444: 'rely on predicted delta CO, the difference: : :.' " Response: It has been modified.

"Line 449: delete 'similar', change 'compared with' to 'comparable to' " Response: It has been modified.

"Line 450: change 'shapefile analysis' to 'shapefiles'" Response: It has been modified.

"Line 452: end sentence as 'from elsewhere in the CONUS domain.' " Response: It has been modified.

"Line 457: change 'outside'to 'bounding' " Response: It has been modified.

"Line 461 – 471: For a structure like this with a colon leading to a list of independent clauses (1-4) (that may or may not contain commas), begin each clause in lower case and terminate all the clauses with a semicolon. Except for the last one, which gets a period." Response: It has been modified.

" Line 463-4: 'to avoid impasse arose by uncertainties' is unclear" Response: It has been changed to "a holistic evaluation approach was adopted so that the fire smoke algorithm was interpreted as a single entity to avoid deadlock due to over-interpretation of uncertainty of the single component in the system;"

"468-9: we were intentionally conservative: : : " Response: It has been modified.

"Line 470: 'outliers' and delete 'sparse' " Response: It has been modified.

"Format used in text for citation is inconsistent." Response: It has been modified.

References:

Chai, T., Kim, H. C., Lee, P., Tong, D., Pan, L., Tang, Y., ... & Stajner, I. (2013). Evaluation of the United States National Air Quality Forecast Capability experimental real-time predictions in 2010 using Air Quality System ozone and NO 2 measurements. Geoscientific Model Development, 6(5), 1831-1850.

Lee, P., McQueen, J., Stajner, I., Huang, J., Pan, L., Tong, D., ... & Lu, S. (2017). NAQFC developmental forecast guidance for fine particulate matter (PM2. 5). Weather and Forecasting, 32(1), 343-360.

Pan, L., Tong, D., Lee, P., Kim, H. C., & Chai, T. (2014). Assessment of NOx and O3 forecasting performances in the US National Air Quality Forecasting Capability before and after the 2012 major emissions updates. Atmospheric environment, 95, 610-619.

Wang, J., Bhattacharjee, P. S., Tallapragada, V., Lu, C. H., Kondragunta, S., da Silva, A., ... & McQueen, J. (2018). The implementation of NEMS GFS Aerosol Component (NGAC) Version 2.0 for global multispecies forecasting at NOAA/NCEP-Part 1: Model descriptions.

Brauer, M., Freedman, G., Frostad, J., Van Donkelaar, A., Martin, R. V., Dentener, F., ... & Balakrishnan, K. (2015). Ambient air pollution exposure estimation for the global burden of disease 2013. Environmental science & technology, 50(1), 79-88.

---

## Author Comment (AC2) · 31 Aug 2019

This article conducts the evaluation the NAQFC simulations including fire smoke particulate (PM25) emission using observations from in-situ, aircraft, and satellite measurements. Several useful indicators/methodologies had been described in this article to identify the signal of fire smoke influence. This article shows valuable information on future evaluation of the impact of fire smoke emission on modeled PM25, as well as the improvement of air quality modeling. However, the manuscript may need major revision to polish its statements for reader to easily understand the message that authors want to deliver. I often found myself taking too much time trying to understand what authors want to say in a paragraph and between paragraphs. This is a common problem of the writing of this manuscript. It lacks transition wording to connect idea between sentences in a paragraph as well as between paragraphs, e.g., the paragraph [lines 461-471] discussed below. I encourage lead author to work closely with co-authors to make the reading easier to deliver the value of this study.

General comments (1) It may be just a personal preference issue, but I suggest authors to rewrite sentence started with "we will compare: : :." or "our simulation: : :" TO "this study will: : :", "the results show: : :", "the comparison between A and B indicates: : :".

Response: It has been modified.

(2) Replace current sentence using "- -" with a complete sentence, e.g., lines 339 and 408.

Response: It has been modified.

(3) Some description belong to figure or table caption and can be removed from main body. It may be easier to understand the main issue, e.g., Lines 323 to 328.

Response: It has been modified.

(4) Avoid adding a single (maybe unrelated) sentence in the middle of a paragraph to stop the flow of message, e.g., line 317 "The ASDTA is a signature identification analysis.". Do not try to clog the article with extra information. Just a few simple and focused descriptions can better deliver your message.

Response: It has been modified.

Specific comments: (1) Lines 76-79: a. The composition of HMS sources are different now from the time this manuscript submitted. To avoid confusion, please add "At the time of this study" at the beginning of the paragraph.

Response: It has been added to text.

b. MODIS and AVHRR is sensors while GOES-12, NASA EOS Aqua, and NOAA-15 : : :etc. are satellites. Please spell out 15/17/18 as NOAA-##. Consider using [: : :...the fire detection from "sensor" on-board "satellite": : :: : :].

Response: Text has been modified.

(2) Lines 240-249: a. How did authors come up with threshold values, i.e., > 20%, < 50%, and < 1? Please provide the reference of the source of the threshold.

Responses: Those threshold values were obtained from this study.

b. Please add "ratio" to the column title of table 2, for columns 9-14.

Response: Table 2 has been modified.

c. My understanding of this paragraph is the ratio should be > 1.2 for EC, OC, and K, < 0.5 for NO3- and SO42-, and < 1 for soil to be classified as "influence by fire smoke". But Table 2 shows NO3- and SO42- ratios at COHU, MACA (two date), and GRSM do not satisfy the criterion, is my understanding wrong? Maybe simply spelling of conditions based on ratio values, such as ratio A > thrershold 1, ratio B < threshold 2, and ratio C >= threshold 3.

Response: If a measurement on IMPROVE site is classified as "influenced by fire smoke", the following conditions must be met at the same time: NO3- and SO42- ratios are less than 1.5; EC, OC and K ratios are greater than 1.2; soil ratio is less than 1.0.

(3) Lines 312-315 a. My knowledge about ASDTA indicates the description of AS-DTA is incorrect. ASDTA uses satellite observed AOD and meteorological fields from

the NCEP operational meteorology model. It does not use HYSPLIT model simulated output. Authors should verify their description with NOAA NESDIS developers of AS-DTA. b. If (a) is correct, please replace all "predicted" ASDTA products with "diagnosis" ASDTA products in manuscript.

Response: Lines 316-331 have been modified.

(4) Lines 341-348 are difficult to understand. My guessing is the authors trying to explain why CMAQ can not capture the fire signal because of (a) do not have a dynamic LBC including the trans-boundary influence of fire smoke PM25 originated from fires outside modeling domain (b) plume rise scheme difference, and (c) different number of fire hotspot used. (c) May not be totally correct, in my opinion, the number of hotspot difference is attributed to difference of domain coverage where HYSPLIT domain is larger. The different model performance between CMAQ and HYSPLIT is already explained by (a), i.e., the HYSPLIT can simulate the long rang transport impact of Canadian fires because it has the fires within its domain.

Response: Lines 350-355 have been modified.

(5) Line 399, the first appearance of "acetonitrile" in this manuscript. Is it CH3CN? Otherwise there is no description in previous paragraphs that this chemical species can be used to identify fire signal.

Response: Acetonitrile is CH3CN. It was defined in Line 352.

(6) Lines 461-471 This paragraph show-up from nowhere and it seems to me has no connection to this study. It is more like a personal experience on the difficulty of fire smoke modeling. I do not know whether items 1-4 are concluded as a result from diagnoses of this study, from a common knowledge of the community, or simply speculation? Since I really have trouble to comprehend the paragraph, I am going to make a bold guess and recommend authors to re-word this paragraph as The comparison of A in this study shows [item 1]. But [item 2] of this study indicates there are other

factors. It is commonly known that [item 3] can impact the results. Thus [item 4] found this study can be used to improve [item 5]. : : :etc.

Response: This manuscript is an evaluation paper. It evaluates the fire algorithm used in real-time operational forecasting. This section introduces our philosophy of evaluating operational models, for example, paying more attention on failed cases and analyzing the reasons. This paragraph has been modified.

(7) Color bar is needed for Figures 7a, 7b, 7d, and 7e, otherwise simple description is needed to let reader know the direction of changing color corresponds to the increase/decrease. Also, those figures are colored-shaded plot. They are not contour plot. The description of figures should be corrected in manuscript.

Response: Graphs have been redrawn.

(8) Figures 9b. Can not see the color of circles for CH3CN concentration.

Response: Figure 9b has been modified.

---

## Author Response (AR2)

*Topical Editor Decision: Publish subject to minor revisions (review by editor) (04 Mar 2020) by Fiona O'Connor*

*Comments to the Author:*

*Dear authors,*

*Thank you for submitting a revised manuscript entitled "Evaluating a fire smoke simulation algorithm in the National Air Quality Forecast Capability (NAQFC) by using multiple observation data sets during the Southeast Nexus (SENEX) field campaign".*

*Having read the revised manuscript and your comments to the reviewers, I am pleased to say that I thought that the writing had improved since the submitted manuscript, making it much clearer to both follow and understand. However, as you'll see from my list of specific comments, there still remains a large number of either poorly constructed sentences or grammatical errors. May I please ask that you correct these before the paper can be accepted for publication?*

*Regards,*

*Fiona O'Connor*

*Specific comments:*

*1. Line 229: Change "A detail of how model" to "Details on how the model"*

Response: did it;

*2. Line 230: Change "follow" to "following"*

Response: did it;

*3. Line 244: Give full names for NO3- and SO42- before using*

Response: did it;

*4. Line 254: Replace "sea salts" with "sea salt"*

Response: did it;

*5. Line 256: Replace "in model simulation" with "in the "with fire" model simulation"*

Response: did it;

**6. Line 258: Give full name for "SSW" before use**

Response: did it;

**7. Line 259: Replace "in MACA" with "at MACA"**

Response: did it;

**8. Line 265: Remove "The model missed fire signal on July 3 at MACA." due to duplication.**

Response: did it;

**9. Line 277 and 278: Replace "represents area" with "represents the area"**

Response: did it;

**10. Line 288: Remove "in terms of resulting" from sentence**

Response: did it;

**11. Line 289: Replace "missed fire emissions" with "missed the fire emissions"**

Response: did it;

**12. Line 293: Replace "in corresponding" with "in the corresponding"**

Response: did it;

**13. Line 300: The whole sentence "The primary reason is that the HYSPLIT smoke simulation is accessed at the invocation of a forecast cycle the HMS fire information which is already one day old due to retrieval latency and cycle-queuing issues" needs re-writing to make clearer.**

Response: it has been changed to "The primary reason is that due to retrieval latency and cycle-queuing problems in HMS, HMS fire information is delayed by one day, which means that HMS today's list can only reflect yesterday's fire information, so HYSPLIT smoke forecasting can only use yesterday's fire information."

**14. Line 303: Update "Huang et al. 2019 (manuscript in preparation)."**

Response: did it;

*15. Line 304: Replace "despite both the HYSPLIT and CMAQ fire plume rise were estimated by the*
*Briggs' equation" with "although the HYSPLIT and CMAQ fire plume rise were both estimated by the*
*Briggs' equation"*

Response: did it;

*16. Line 306: Replace "at daytime" with "during daytime"*

Response: did it;

*17. Line 318: Replace "ASDTA is originally generate to provide operational support for verification of*
*the NOAA HYSPLIT dispersion model predicts smoke plume direction and extension" with "ASDTA,*
*originally generated to provide operational support for verification of the NOAA HYSPLIT dispersion*
*model, predicts smoke plume direction and extension"*

Response: did it;

*18. Line 337: Replace "in observation" with "in the observations"*

Response: did it;

*19. Line 340: Change "Differences were attributable to:" to "Differences were attributable to a number*
*of reasons: "*

Response: did it;

*20. Line 357: Change "evaluations" to "evaluation" and change "wind field" to "wind fields"*

Response: did it;

*21. Line 360: Change "An additional uncertainty arose in the difference of temporal resolutions" to*
*"An additional uncertainty arose due to the difference in temporal resolution"*

Response: did it;

*22. Line 363; Change "flight transects" to "a flight transect"*

Response: did it;

***23. Line 365: Change "color of flight path" to "color along the flight path"***

Response: did it;

***24. Line 366: Change "from the surface" to "from surface"***

Response: did it;

***25. Line 370: Change "Model simulation" to "The model simulation"***

Response: did it;

***26. Line370: Change "long range transports" to "long range transport"***

Response: did it;

***27. Line 372: Change "observation" to "concentrations"***

Response: did it;

***28. Line 373: Change "or smoke plume around" to "or smoke plumes around"***

Response: did it;

***29. Line 382: Change "all of evaluation" to "all of the evaluation"***

Response: did it;

***30. Line 383: Change "reproduce fire" to "reproduce the fire"***

Response: did it;

***31. Line 389: Change "concentration" to "concentrations"***

Response: did it;

*32. Line 403: Change "Fire signals showed substantial influences on aircraft measurement" to "Fire signals have a substantial influence on the aircraft measurements"*

Response: did it;

*33. Line 411: Change "were" to "are"*

Response: did it;

*34. Line 413: Change "with CH3CN measured concentration above 400 ppt" to "with observed CH3CN concentrations above 400 ppt"*

Response: did it;

*35. Line 418: Change "Other explanation was from Fig. 11b, which illustrated hotspots in hmx.txt." to "Another explanation can be seen from Fig. 11b, which illustrates hotspots in hmx.txt."*

Response: did it;

*36. Line 420: Change "spots by HMS before quality control were showed" to "spot by HMS before quality control are shown"*

Response: did it;

*37. Line 420: Change "there were clusters" to "there are clusters"*

Response: did it;

*38. Line 422: Change "In most of time" to "In most cases"*

Response: did it;

*39. Line 424: Change "For this case, there seem to have been thin clouds overhead and thicker clouds in the vicinity" to "For this particular case, there seem to have been thin clouds overhead and thicker clouds in the vicinity"*

Response: did it;

**40. Line 427: Please add a full stop at the end of the sentence**

Response: did it;

**41. Line 436: Change "For IMPROVE data" to "For the IMPROVE data"**

Response: did it;

**42. Line 438: Change "by CMAQ" to "by the CMAQ"**

Response: did it;

**43. Line 440: Change "overlapping" to "overlap"**

Response: did it;

**44. Line 451: Change "inflows" to "inflow"**

Response: did it;

**45. Line 622: Change "shown along flight transect" to "shown along a flight transect"**

Response: did it;

**46. Line 644: Change "highlighted" to "highlight"**

Response: did it;

**47. Line 651: Change "a backward trajectory analysis for CH3CN concentration in ppt greater" to "A backward trajectory analysis for CH3CN concentrations greater"**

Response: did it;

[revised manuscript text omitted]